# A NIN-LIKE PROTEIN mediates nitrate-induced control of root nodule symbiosis in *Lotus japonicus*

Hanna Nishida[1,2,3], Sachiko Tanaka[1], Yoshihiro Handa[1], Momoyo Ito[3], Yuki Sakamoto[4], Sachihiro Matsunaga[4,5], Shigeyuki Betsuyaku[3], Kenji Miura[3], Takashi Soyano[1,2], Masayoshi Kawaguchi[1,2] & Takuya Suzaki[3]

Legumes and rhizobia establish symbiosis in root nodules. To balance the gains and costs associated with the symbiosis, plants have developed two strategies for adapting to nitrogen availability in the soil: plants can regulate nodule number and/or stop the development or function of nodules. Although the former is accounted for by autoregulation of nodulation, a form of systemic long-range signaling, the latter strategy remains largely enigmatic. Here, we show that the *Lotus japonicus NITRATE UNRESPONSIVE SYMBIOSIS 1* (*NRSYM1*) gene encoding a NIN-LIKE PROTEIN transcription factor acts as a key regulator in the nitrate-induced pleiotropic control of root nodule symbiosis. NRSYM1 accumulates in the nucleus in response to nitrate and directly regulates the production of CLE-RS2, a root-derived mobile peptide that acts as a negative regulator of nodule number. Our data provide the genetic basis for how plants respond to the nitrogen environment and control symbiosis to achieve proper plant growth.

[1] National Institute for Basic Biology, Okazaki, Aichi, Japan. [2] School of Life Science, SOKENDAI (The Graduate University for Advanced Studies), Okazaki, Aichi, Japan. [3] Graduate School of Life and Environmental Sciences, University of Tsukuba, Tsukuba, Ibaraki, Japan. [4] Imaging Frontier Center, Organization for Research Advancement, Tokyo University of Science, Noda, Chiba, Japan. [5] Department of Applied Biological Science, Faculty of Science and Technology, Tokyo University of Science, Noda, Chiba, Japan. Correspondence and requests for materials should be addressed to T.S. (email: suzaki.takuya.fn@u.tsukuba.ac.jp)

In a nitrogen-deficient environment, legumes can form specialized symbiotic organs, root nodules, through association with rhizobia. Root nodules enable plants to obtain a nitrogen source fixed from atmospheric nitrogen. To establish the root nodule symbiosis, a sequential progression of several key processes needs to occur in the root. Upon the perception of a signal from rhizobia, plants form intracellular tube-like structures called infection threads that are used to accommodate rhizobia within the host cells. Simultaneously, dedifferentiation of the cortical root cells is induced, and these cells proliferate to form nodule primordia. During the course of nodule development, rhizobia are endocytosed into the nodule cells and are able to fix nitrogen[1,2]. Owing to the symbiosis, legumes can grow in soil without a nitrogen source; however, the symbiosis is known to be an energy-consuming activity in which photosynthates are used as an energy source to drive processes such as cortical cell proliferation and nitrogen fixation[1,3]. Therefore, to optimize their growth, plants need to maintain a balance of gains and costs; that is, the nitrogen demands of plants must be fulfilled without unnecessary loss of carbon. To this end, plants have developed two major ways to negatively regulate the symbiosis.

First, legumes control the number of nodules per root system through a mechanism called autoregulation of nodulation (AON), a systemic long-range signaling between roots and shoots[4–6]. In the model legume *Lotus japonicus*, expression of three *CLA-VATA3/ESR-related* (*CLE*) genes, *CLE-ROOT SIGNAL 1* (*CLE-RS1*), *-RS2* and *-RS3* is induced by rhizobial infection of the roots[7,8]. The resulting CLE-RS1/2/3 peptides, presumably root-derived mobile signals, negatively affect nodulation and may interact with a shoot-acting leucine-rich repeat receptor-like kinase (LRR-RLK) named HYPERNODULATION ABERRANT ROOT FORMATION 1 (HAR1) that is proposed to form a receptor complex with other LRR-RLK, KLAVIER (KLV)[9] and LRR-RL protein, LjCLV2[10]. As a result, the production of secondary shoot-derived signals is induced, and these signals are transported down to the root to block further nodule development[8,11–13]. Loss-of-function mutations in any gene involved in the AON commonly result in deficient plant growth due to the formation of an excess number of nodules[14–16], demonstrating the importance of maintaining a symbiotic balance through AON. Systemic negative feedback control appears to have a conserved molecular mechanism among leguminous species, as functional counterparts of HAR1 and CLE-RS1/2/3 have been identified in other legumes such as *Medicago truncatula* and *Glycine max*[17–20].

Second, plants have the ability to control root nodule symbiosis in response to nitrogen availability in the soil. Plants may cease the symbiosis if there is a sufficient nitrogen source available in their environment, thereby enabling plants to save the cost associated with nodulation. In this context, plants can regulate each of the multiple phases of root nodule symbiosis, including rhizobial infection, nodule initiation, nodule growth, and nitrogen fixation activity, in response to nitrate, a major form of inorganic nitrogen in soil[21,22]. High nitrate is also known to accelerate nodule senescence or disintegration[23]. In addition to their hypernodulating phenotypes, mutations in key LRR-RLKs involved in the AON in several legumes, such as *L. japonicus* HAR1 and KLV, *M. truncatula* SUPER NUMERIC NODULES and *G. max* NODULE AUTOREGULATION RECEPTOR KINASE, retain nodule formation even in the presence of a high nitrate concentration[14,15,17,24]. Furthermore, expression of the *CLE-RS2*, *-RS3*, and *LjCLE40* genes is induced not only by rhizobial infection but also by nitrate application[8]. These observations suggest that the mechanism for nitrate-induced control of nodulation shares common elements with the AON[7]. In contrast, some findings suggest that fundamental knowledge of AON is insufficient to account for a pleiotropic regulatory mechanism[25,26], indicating that new factors await discovery.

In this study, we identify a novel *L. japonicus* mutant, *nitrate unresponsive symbiosis 1* (*nrsym1*). The *nrsym1* mutants are unable to cease root nodule symbiosis under nitrate-sufficient conditions. Our results show that *NRSYM1* encodes a NIN-LIKE PROTEIN (NLP) transcription factor and mediates nitrate-induced pleiotropic control of root nodule symbiosis. In addition, we determine the specific role of AON components in this process. That is, NRSYM1 directly regulates *CLE-RS2* expression in response to nitrate, thereby triggering the negative regulation of nodule number.

## Results

**NRSYM1 mediates the nitrate-induced control of nodulation.** To elucidate the genetic mechanism relevant to the nitrate-induced control of root nodule symbiosis, we screened for mutants involved in the nitrate response during nodulation using ethylmethane sulfonate (EMS)-treated *L. japonicus* wild-type (WT) MG-20 plants. Two allelic recessive mutants named *nitrate unresponsive symbiosis 1-1* (*nrsym1-1*) and *nrsym1-2* were identified. F1 plants derived from a cross between *nrsym1-1* and the WT MG-20 parental line normally responded to nitrate. In the F2 population, nitrate-sensitive and nitrate-tolerant plants segregated in an ~3:1 ratio (17 nitrate-sensitive and 7 nitrate-tolerant plants). These results indicate that the *nrsym1* mutation is inherited as a recessive trait. The *nrsym1-1* mutants exhibited normal nodulation under nitrate-free conditions. Although 10 mM nitrate significantly attenuated nodulation in WT, the *nrsym1-1* plants formed mature nodules in the presence of a high nitrate concentration (Fig. 1a). To establish root nodule symbiosis, a sequence of key processes, including nodule initiation, rhizobial infection, nodule growth, and nitrogen fixation activity, are essential and are under nitrate control[21,22]. The nodule number of WT gradually decreased with increasing concentrations of nitrate, and the formation of small and immature nodules suggested that premature arrest of nodule development had occurred. In contrast, in the *nrsym1-1* mutant, nodule number was primarily normal and mature nodules formed even in the presence of 10 mM nitrate. Under 50 mM nitrate conditions, nodulation was attenuated even in the *nrsym1-1* mutants (Fig. 1b). In WT, the number of infection threads, an indicator of rhizobial infection foci, was significantly reduced by nitrate, but the nitrate-induced reduction of infection thread number was not observed in the *nrsym1-1* mutants (Fig. 1c). Next, to focus on the effect of nitrate on nodule growth, plants were first grown with rhizobia on nitrate-free agar plates. After 7 days, by which time nodule primordia had formed, the plants were transferred to new agar plates containing 0 or 10 mM nitrate, and nodule sizes were measured every 5 days. Whereas WT nodule size under the nitrate-free condition increased with time, 10 mM nitrate arrested nodule growth. In contrast, in the *nrsym1-1* mutant, nodule growth was not affected by high nitrate (Fig. 1d). Finally, the effect of nitrate on the nitrogen fixation activity of nodules was investigated (Fig. 1e). Plants were grown with rhizobia in the absence of nitrate for 21 days, by which time mature nodules had formed. Then, 0 or 10 mM nitrate was supplied to the plants. After 3 days, the acetylene reduction activity (ARA) of nodules was measured for each plant. In WT, nitrate significantly reduced the ARA of nodules. In contrast, the inhibitory effect was not observed in the *nrsym1-1* mutants. The *nrsym1-2* mutants had a nitrate-tolerant phenotype similar to *nrsym1-1* (Figs. 1a, b; Supplementary Fig. 1a–c). These data indicate that the *nrsym1* mutation eliminates the pleiotropic nitrate-induced inhibition of root nodule symbiosis.

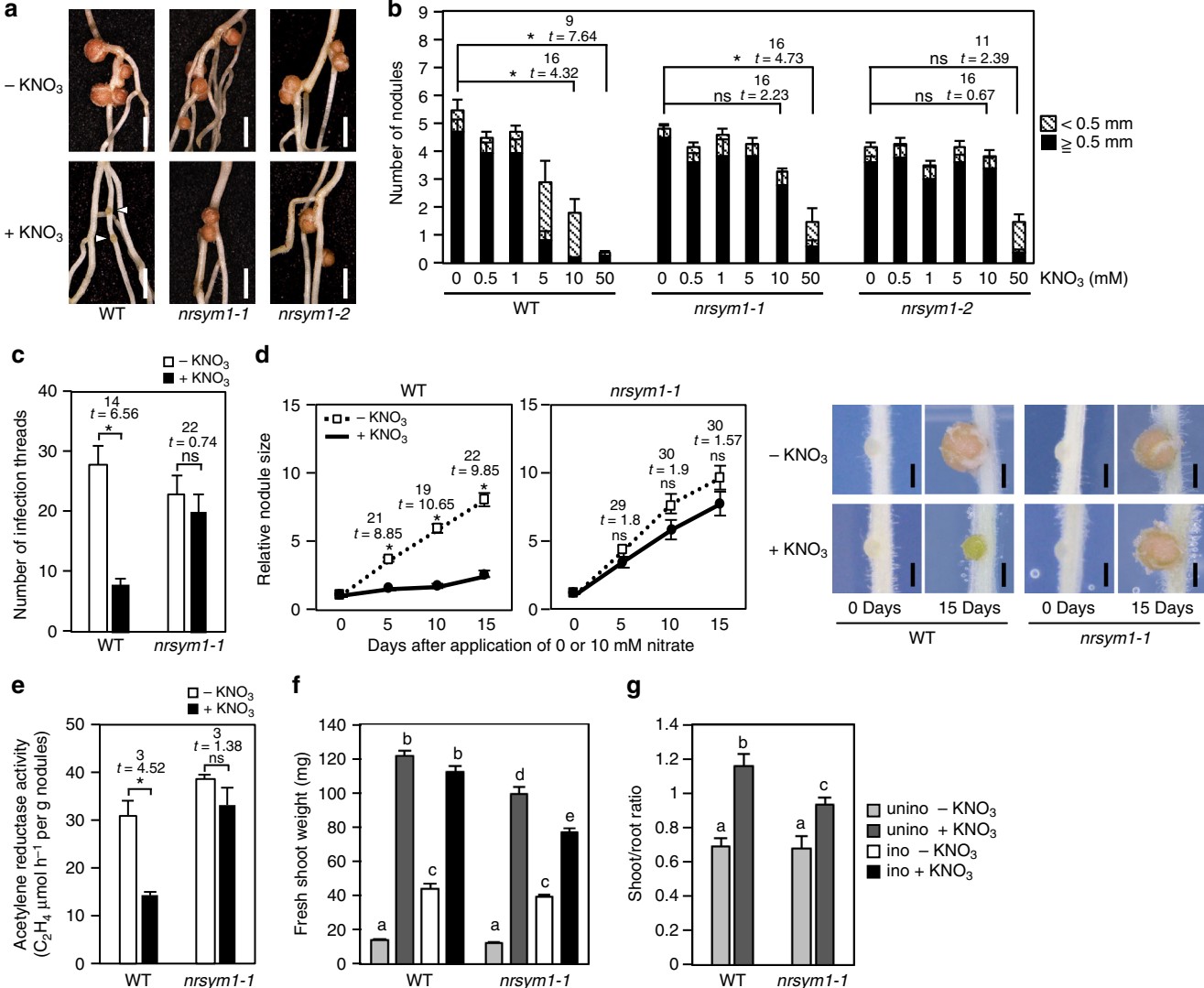

**Fig. 1** The effect of the *nrsym1* mutation on nodulation and plant growth. **a** Nodule phenotypes of WT, the *nrsym1-1* mutant, and the *nrsym1-2* mutant treated with 0 or 10 mM $KNO_3$ at 21 days after inoculation (dai). Arrowheads indicate small and premature nodules. Scale bars: 2 mm. **b** The number of nodules in WT, the *nrsym1-1* mutants, and the *nrsym1-2* mutants in the presence of different concentrations of $KNO_3$ (0–50 mM) at 21 dai (*n* = 9 plants). **c** The number of infection threads in WT and the *nrsym1-1* mutants with 0 or 10 mM $KNO_3$ at 7 dai with rhizobia that constitutively express *LacZ* (*n* = 12 plants). **d** Relative nodule size (daily nodule size/nodule size on day 0) of WT and the *nrsym1-1* mutants (*n* = 13–19 nodules). Individual nodule size was measured at 0, 5, 10, and 15 days after the transfer to agar plates with 0 or 10 mM $KNO_3$. *$P < 0.05$ (Student's *t*-test compared 0 mM $KNO_3$-treated nodules with 10 mM $KNO_3$-treated nodules on the same day). ns, not significant. Scale bars: 0.5 mm. **e** Acetylene reduction activity (ARA; μmol h$^{-1}$ per g nodules) of nodules formed on WT and the *nrsym1-1* mutants (*n* = 4 plants). Twenty-one dai plants without $KNO_3$ were supplied with 0 or 10 mM $KNO_3$, and after 3 days the ARA of nodules from each plant was measured. **f** Fresh shoot weight and (**g**) shoot to root fresh weight ratio of WT and the *nrsym1-1* mutants grown in 0 or 10 mM $KNO_3$ on 21 dai (ino) or without rhizobia (unino; *n* = 10–12 plants). Error bars indicate SEM. *$P < 0.05$ by Student's *t*-test. ns, not significant. Degrees of freedom are shown above the *t*-values (**b–e**). Columns with the same lower-case letter indicate no significant difference (Tukey's test, $P < 0.05$; **f**, **g**)

We next examined the effects of the *nrsym1* mutation on plant growth. Whereas nitrate promoted shoot growth of both WT and *nrsym1-1*, in the absence of rhizobia *nrsym1-1* had a smaller fresh shoot weight than WT (Fig. 1f). The shoot–root fresh weight ratio, a representative marker for nutrient starvation status[27], was lower in *nrsym1-1* compared with WT under nitrate-sufficient conditions (Fig. 1g). Thus, in addition to its symbiotic roles, NRSYM1 seems to function in non-symbiotic nitrate-related processes. In the presence of nitrate, the shoot weight of inoculated WT was indistinguishable from that of uninoculated WT plants. Of note, simultaneous application of rhizobia and nitrate to *nrsym1-1* caused about a 22% reduction in shoot growth compared with uninoculated and nitrate-treated *nrsym1-1* plants (Fig. 1f). The *nrsym1-2* mutants had a shoot phenotype

similar to that of *nrsym1-1* (Supplementary Fig. 1d, e). These results indicate that nodulation in a nitrate-sufficient condition can be harmful to plant growth.

In addition to nitrate, plants are known to control root nodule symbiosis in response to ammonium[28]. We then analyzed nodulation of *nrsym1* under ammonium-sufficient conditions (Supplementary Fig. 1f). In the presence of 10 mM ammonium, nodulation of *nrsym1* and WT was attenuated. This result suggests that NRSYM1 is not involved in the ammonium-induced control of nodulation.

**High nitrate conditions reduce nodule cell size.** To characterize the effect of nitrate on nodule growth in more detail, we observed

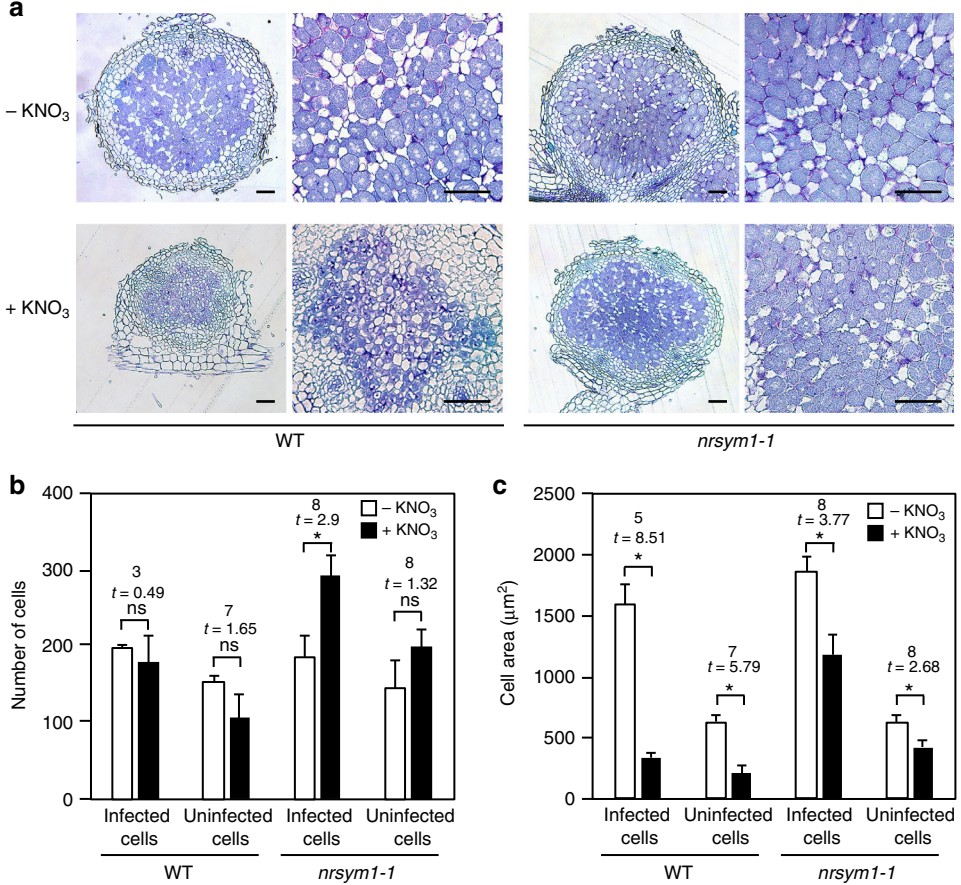

**Fig. 2** The effect of nitrate on nodule growth. **a** Sections through 21 dai nodules of WT and the *nrsym1-1* mutants grown in the presence of 0 or 10 mM KNO$_3$. Sections were stained with toluidine blue. Scale bars: 100 μm. **b** The number of cells and (**c**) the cell area (μm$^2$) in the inner region of nodule sections. After images of individual nodule sections of maximum diameter had been collected, cell number and cell area were measured using ImageJ software (http://imagej.nih.gov/ij/; *n* = 4–5 nodule section). Infected and uninfected cells located in the inner region of nodule parenchyma were scored for the presence or absence of cell staining, respectively. To calculate cell area, the area of all infected and uninfected cells was measured and averaged (170–637 cells per nodule section). Using the obtained average cell area in respective nodule sections, the average cell area per nodule section was calculated. *$P$ < 0.05 by Student's *t*-test. ns, not significant. Degrees of freedom are shown above the *t*-values. Error bars indicate SEM

nodule sections of WT and the *nrsym1-1* mutants grown in the presence of 0 or 10 mM nitrate. WT nodules formed in the presence of 10 mM nitrate were obviously small (Fig. 1a). Examination of WT nodule sections from plants grown in nitrate revealed that rhizobia-colonized infected cells were recognizable, but their size was much smaller compared with the WT nitrate-free nodules (Fig. 2a). We then measured the number and size of cells located in the inner region of nodule parenchyma. In WT nodules grown in the presence of nitrate, the number of cells was comparable to that of nitrate-free nodules (Fig. 2b). On the other hand, nitrate reduced the cell sizes of both infected and uninfected cells (Fig. 2c), suggesting that the reduction in nitrate-induced nodule size was due to a smaller cell size rather than cell number. Sections from *nrsym1-1* nodules developed in the presence of nitrate were largely indistinguishable from the *nrsym1-1* nodules developed in the absence of nitrate (Fig. 2a). Although nitrate reduced the cell sizes of both infected and uninfected *nrsym1-1* cells, the number of infected cells increased (Figs. 2b, c).

**NRSYM1 encodes an NLP transcription factor.** We first sought to isolate *NRSYM1* by a map-based cloning approach. The *NRSYM1* locus was mapped to a region between two simple sequence repeat markers, TM1417 and TM0366, on chromosome

5 (Supplementary Fig. 2). Subsequently, a genome-resequencing approach using the *nrsym1-1* and *nrsym1-2* mutants (Supplementary Table 1) identified point mutations in the gene, chr5. CM0148.170.r2.a, which was previously identified as *LjNLP4*[29]. There are two nucleotide substitutions in the gene: a C-to-T substitution causing the formation of a stop codon Q327STOP (*nrsym1-1*) and a G-to-A substitution causing the replacement of valine by isoleucine V283I (*nrsym1-2*; Fig. 3a, b). A 7.5-kb genomic fragment encompassing the entire *NRSYM1* locus was introduced into the *nrsym1-1* mutants by *Agrobacterium rhizogenes*-mediated hairy root transformation. The mutant roots carrying the complementation construct normally responded to nitrate (Fig. 3c), indicating that the *nrsym1* phenotype results from mutation of the gene. *NRSYM1* encodes a protein with sequence similarity to Arabidopsis NLP transcription factors[30]. In Arabidopsis, NLPs have a role as master regulators of nitrate-inducible gene expression[31]. Phylogenetic analysis showed that NRSYM1 belongs to a clade containing AtNLP6 and AtNLP7 (Supplementary Fig. 3). Constitutive expression of *AtNLP6* or *AtNLP7* by the *LjUBQ* promoter partially rescued the nodule number phenotype of *nrsym1-1*. On the other hand, mature nodules still formed on the roots (Supplementary Fig. 4). These results suggest that the functions of NRSYM1 and AtNLP6/7 are partially conserved. NRSYM1 consists of an N-terminal

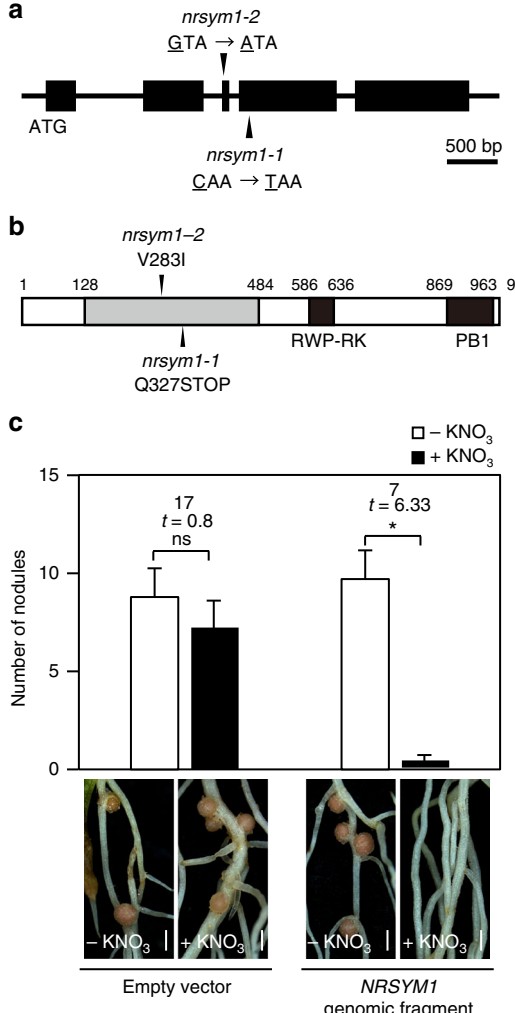

**Fig. 3** Structure of the *NRSYM1* gene. **a** Exon–intron structure of the *NRSYM1* gene. Black boxes indicate exons. Arrowheads indicate locations of the *nrsym1* mutations. **b** Structure of the deduced NRSYM1 protein. An N-terminal conserved region is indicated by the gray box. C-terminal regions containing an RWP-RK DNA-binding domain (RWP-RK) and a conserved domain (PB1) are shown as black boxes. Arrowheads indicate locations of the *nrsym1* mutations. **c** Complementation of the *nrsym1* nodulation phenotype. Representative transgenic hairy roots of *L. japonicus* carrying a control empty vector or a 7.5-kb genomic fragment encompassing the entire *NRSYM1* locus. Transgenic roots were identified by GFP fluorescence. The number of nodules was counted in the transgenic roots of *nrsym1-1* mutants containing either an empty vector or a genomic fragment of *NRSYM1* in the presence of 0 or 10 mM KNO₃ at 21 dai ($n = 8$–10 plants). *$P < 0.05$ by Student's *t*-test. ns, not significant. Degrees of freedom are shown above the *t*-values. Error bars indicate SEM. Scale bars: 1 mm

conserved domain, an RWP-RK DNA-binding domain, and a conserved PB1 domain (Fig. 3b). Placement of a premature stop codon in the N-terminal conserved domain of *nrsym1-1* implies that *nrsym1-1* is a null mutant. The amino-acid residue (V283) that is mutated in *nrsym1-2* is highly conserved among NLPs (Supplementary Fig. 5) and the severity of the *nrsym1-2* mutation is similar to that of *nrsym1-1* (Fig. 1a, b; Supplementary Fig. 1a–e), suggesting that V283 may be a critical amino-acid residue for NRSYM1 function.

We then analyzed the expression pattern of *NRSYM1* in some vegetative and reproductive organs by real-time RT-PCR. *NRSYM1* expression was widely observed in the organs examined

(Supplementary Fig. 6a). The level of *NRSYM1* expression was unchanged by nitrate treatment and seemed to be unaffected by rhizobial inoculation, at least at the whole-root level (Supplementary Fig. 6a,b). Reporter gene analysis using a *pNRSYM1:GUS* construct showed that *NRSYM1* was expressed in the root vascular tissue (Supplementary Fig. 6c), nodule primordia (Supplementary Fig. 6d), and mature nodules (Supplementary Fig. 6e).

**HAR1 regulates the nitrate-induced control of nodule number.** Several observations suggest that AON is implicated in nitrate-induced control of root nodule symbiosis[15,28]. To assess the potential genetic interaction between NRSYM1 and HAR1, a key regulator of AON, we analyzed the *nrsym1 har1* double-mutant phenotype (Figs. 4a–d). Under nitrate-free and -sufficient conditions, *nrsym1-1 har1-7* mutants formed an excess number of nodules that were similar to those of the *har1-7* mutants (Fig. 4b). In the presence of 10 mM nitrate, however, the *har1-7* mutants continued to produce an elevated number of small, white immature nodules (Fig. 4b) similar to those formed in WT in the presence of high nitrate. In *har1-7*, infection thread number and nodule size were significantly reduced by nitrate (Figs. 4a, c). In contrast, the nitrate-induced reduction in infection thread number and nodule size was masked by the presence of the *nrsym1* mutation (Figs. 4a, c). In the *har1-7* mutants, nitrate reduced the nitrogen fixation activity, but the *nrsym1-1 har1-7* mutants were tolerant of the reduction (Fig. 4d). Hence, these results indicate that HAR1 may be involved in the regulation of nitrate-induced inhibition of nodule number, but rhizobial infection, nodule growth, and nitrogen fixation activity are controlled through a mechanism independent of HAR1.

Reciprocal grafting experiments were then performed using WT, *nrsym1-1*, and *har1-7* (Fig. 4e). Whereas WT(scion)/*nrsym1-1*(rootstock)-grafted plants showed a nitrate-tolerant phenotype in the presence of 10 mM nitrate, nodulation of *nrsym1-1*/WT-grafted plants was nearly eliminated. Thus, root-acting NRSYM1 seems to have a role in the nitrate-induced control of nodulation. HAR1 was previously shown to act in the shoot[12]. Whereas *nrsym1-1*/*har1-7*-grafted plants exhibited reduced nodulation in the presence of nitrate, *har1-7*/*nrsym1-1*-grafted plants had a nodule number similar to *har1-7*/*har1-7*-grafted plants. In the *har1-7*/*nrsym1-1*-grafted plants, mature nodules formed in the presence of high nitrate but were rarely observed in the *har1-7*/*har1-7*-grafted plants.

**NRSYM1 controls nitrate-inducible gene expression.** The expression of *CLE-RS2* is induced not only by rhizobia inoculation but also by nitrate application[7]. To gain insight into the role of NRSYM1 in the nitrate response, we investigated the expression of a nitrate-inducible symbiotic gene (*CLE-RS2*) and two non-symbiotic genes (*NITRATE REDUCTASE* (*NIA*) and *NITRITE REDUCTASE 1* (*NIR1*)) by real-time RT-PCR. *LjNIA* and *LjNIR1*, respectively, encode nitrate and nitrite reductase involved in nitrate assimilation[32,33]. The expression of *CLE-RS2*, *LjNIA*, and *LjNIR1* was strongly induced in WT by a 24-h nitrate treatment, but the induction levels were much lower in *nrsym1-1* roots (Figs. 5a–c). The *nrsym1-2* mutants had a defect similar to that in *nrsym1-1* (Supplementary Fig. 7a–c). We next examined nitrate-inducible gene expression at shorter time points (Supplementary Fig. 7d–f). Whereas the *LjNIA* and *LjNIR1* genes in WT were upregulated 30 min after nitrate treatment, the induction of *CLE-RS2* was detectable 6 h after nitrate treatment. The induction levels of these genes in *nrsym1-1* roots were lower than those of WT at all time points. In addition, we examined the effects of lower concentrations of nitrate that do not inhibit

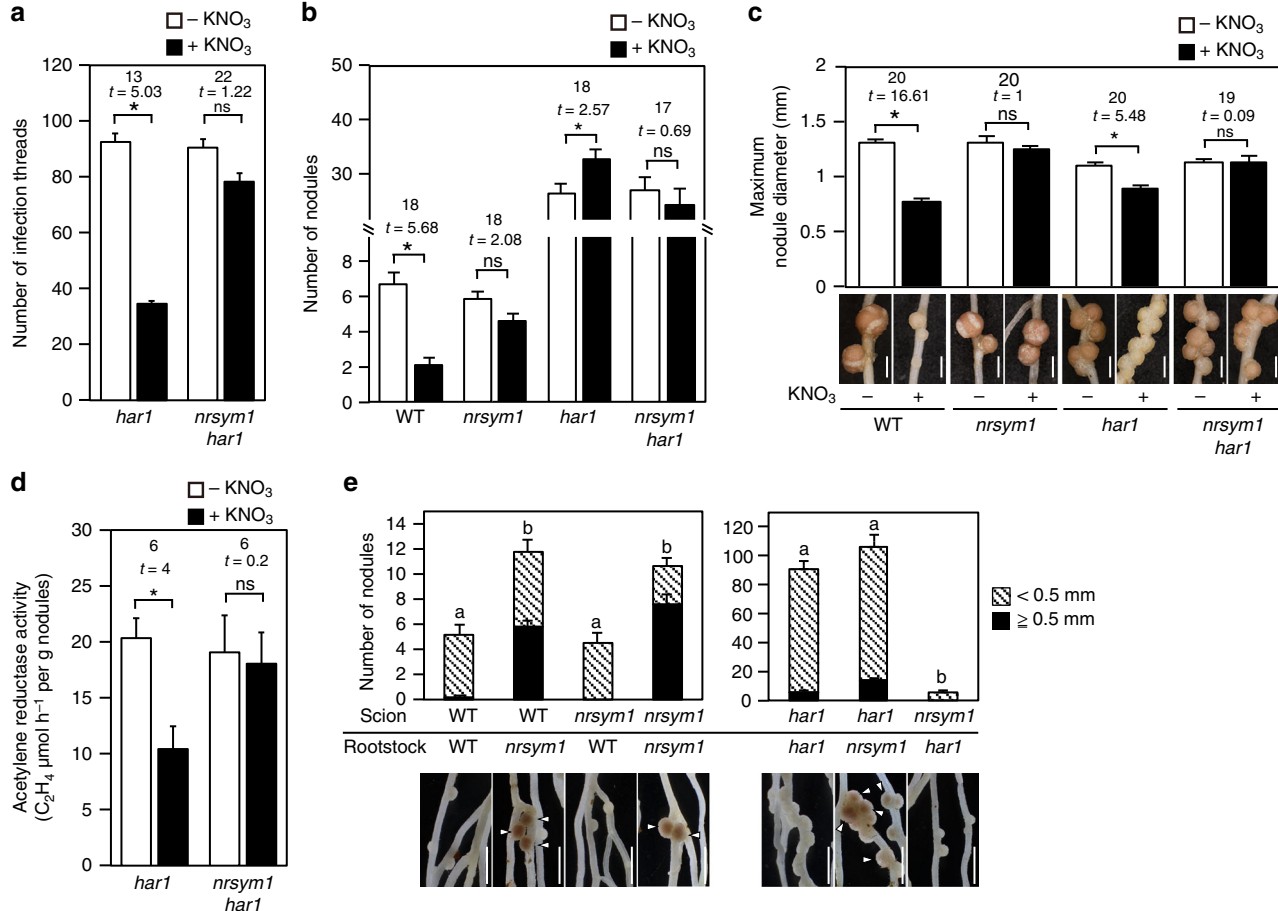

**Fig. 4** Effects of nitrate on nodulation in the *nrsym1 har1* double mutants. **a** The number of infection threads in the *har1-7* mutants and the *nrsym1-1 har1-7* double mutants (*n* = 11–12 plants). Plant growth conditions were the same as those shown in Fig. 1c. **b** The number of nodules and (**c**) maximum nodule diameter (mm) in WT, the *nrsym1-1* mutants, the *har1-7* mutants, and the *nrsym1-1 har1-7* double mutants grown in the presence of 0 or 10 mM KNO$_3$ at 21 dai (*n* = 9–12 plants). **d** Acetylene reduction activity (ARA; μmol h$^{-1}$ per g nodule) of nodules formed on the *har1-7* mutants and the *nrsym1-1 har1-7* double mutants (*n* = 4 plants). Plant growth conditions were the same as those shown in Fig. 1e. **e** Nodulation and nodule numbers of plants derived from shoot–root grafts having WT, *nrsym1-1*, and *har1-7* genotypes. Plants were grown in the presence of 10 mM KNO$_3$ for 21 dai (*n* = 13 plants). Arrowheads indicate mature nodules. *$P < 0.05$ by Student's *t*-test. ns, not significant. Degrees of freedom are shown above the *t*-values (**a–d**). Columns with the same lower-case letter indicate no significant difference (Tukey's test, $P < 0.05$; **e**). Error bars indicate SEM. Scale bars: 1 mm (**c**); 2 mm (**e**)

nodulation on gene expression. Application of 200 μM nitrate induced *CLE-RS2*, *LjNIA*, and *LjNIR1* expression in a NRSYM1-dependent manner (Supplementary Fig. 7g–i). Whereas *LjNIA* and *LjNIR1* induction by nitrate was dependent on having NRSYM1 in roots (Fig. 5b, c), NRSYM1 was not required for inducing expression of the two genes in leaves (Supplementary Fig. 7j, k). Next, the effect of *NRSYM1* constitutive expression on the expression of these genes was investigated. Under nitrate-free conditions, all the tested genes had similar expression levels in hairy roots transformed with control *pLjUBQ:GUS* or *pLjUBQ:NRSYM1* constructs (Fig. 5e–h). In contrast, nitrate application significantly activated *CLE-RS2*, *LjNIA*, and *LjNIR1* in the roots that constitutively expressed *NRSYM1* (Fig. 5e–g), suggesting that NRSYM1 plays a role in the regulation of *CLE-RS2*, *LjNIA,* and *LjNIR1* expression in response to nitrate. Nitrate application, *nrsym1* mutation, nor *NRSYM1* expression affected *CLE-RS1* expression (Fig. 5d, h).

NODULE INCEPTION (NIN) is a key transcription factor that regulates nodule organogenesis[34–37]. The expression of *NIN* tended to be downregulated by nitrate in WT as previously shown[28], and the effect was not observed in *nrsym1-1* roots (Supplementary Fig. 7l).

**NRSYM1 directly regulates *CLE-RS2* and *LjNIR1* upon nitrate.**
An observation of the *NRSYM1*-dependent expression of some nitrate-inducible genes (Fig. 5a–c, e–g) led us to postulate that NRSYM1, an NLP transcription factor, directly regulates the expression of these genes. We then investigated whether these genes are direct targets of NRSYM1 by chromatin immunoprecipitation (ChIP)-qPCR analysis using transgenic hairy roots carrying the *pLjUBQ:NRSYM1-myc* construct. As the *nrsym1-1* mutant roots carrying the construct normally responded to nitrate, the NRSYM1-myc translational fusion seemed to be functional (Supplementary Fig. 8a). NRSYM1 and NIN belong to the NLP family whose members are characterized by the presence of a conserved RWP-RK DNA-binding domain[29,30]. In Arabidopsis, NLPs bind to the nitrate-responsive *cis*-element (NRE) of a number of nitrate-inducible genes[31]. Recently, NIN was shown to bind to the so-called NIN-binding nucleotide sequence (NBS) in the *CLE-RS2* and *LjNIR1* promoters, a sequence that is structurally similar to the NRE[38,39] (Supplementary Fig. 8b). Thus, we designed primer sets in the *CLE-RS2* and *LjNIR1* promoter region based on the presence of NRE/NBS (regions 1, 3, 4 of the *CLE-RS2* promoter and region 3 of the *LjNIR1* promoter) and used them for qPCR analysis after ChIP (Fig. 6a). Chromatin

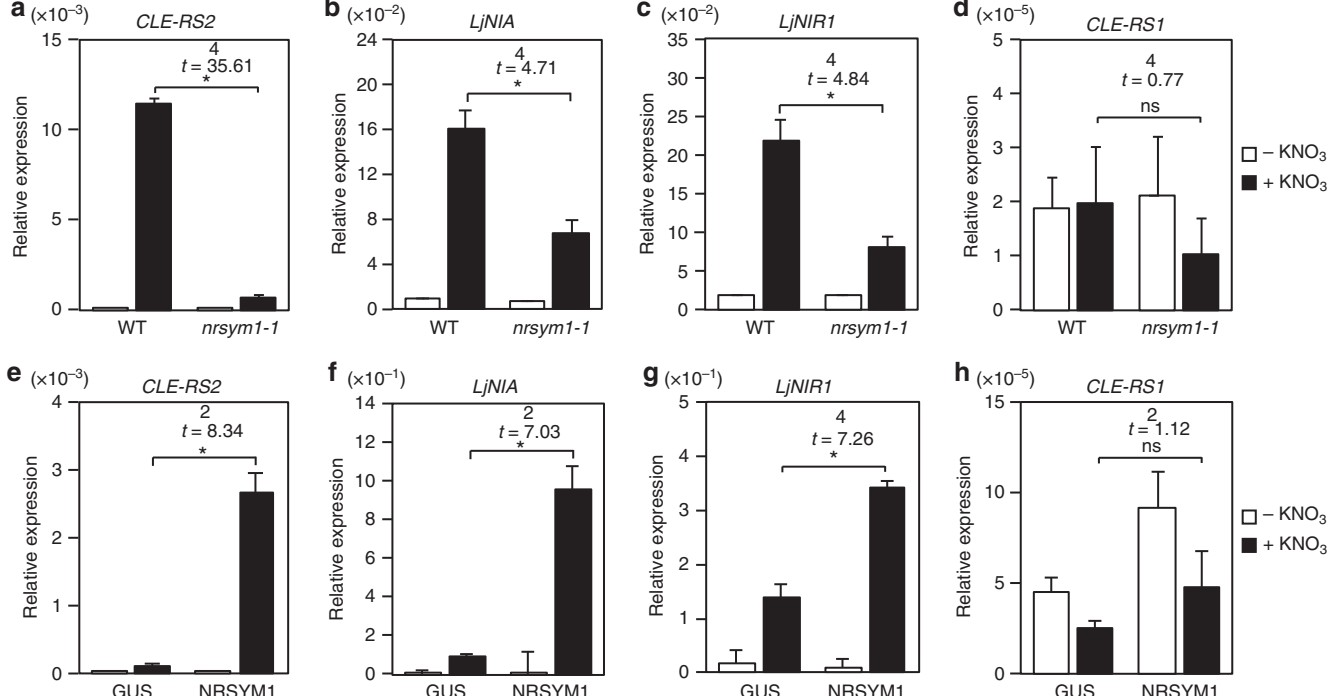

**Fig. 5** The effect of the *nrsym1* mutation or *NRSYM1* constitutive expression on gene expression. **a–d** Real-time RT-PCR analysis of (**a**) *CLE-RS2*, (**b**) *LjNIA*, (**c**) *LjNIR1*, and (**d**) *CLE-RS1* expression in roots of WT and the *nrsym1-1* mutants. **e–h** Real-time RT-PCR analysis of (**e**) *CLE-RS2*, (**f**) *LjNIA*, (**g**) *LjNIR1*, and (**h**) *CLE-RS1* expression in transgenic hairy roots produced from WT containing the *pLjUBQ:GUS* or *pLjUBQ:NRSYM1* constructs. Each cDNA sample was prepared from total RNA derived from an uninoculated whorl of roots grown in the presence of 0 or 10 mM $KNO_3$ for 24 h. The expression of *LjUBQ* was used as the reference. Error bars indicate SEM ($n = 3$ independent pools of roots). *$P < 0.05$ by Student's *t*-test. ns, not significant. Degrees of freedom are shown above the *t*-values

suspensions were prepared from *pLjUBQ:NRSYM1-myc* roots that were incubated with 10 mM nitrate or without nitrate for 24 h, and ChIP was performed with polyclonal anti-myc antibodies. The *CLE-RS2* promoter region 1 was significantly enriched in nitrate-treated roots compared with that of roots receiving no nitrate (Fig. 6a, b). *LjNIR1* promoter region 3 was also enriched in nitrate-treated roots (Fig. 6a, c). These results suggest that NRSYM1 can directly bind to the promoter regions of *CLE-RS2* and *LjNIR1* in a nitrate-dependent manner.

To verify physical interaction between NRSYM1 and NRE/NBS of the *CLE-RS2* and *LjNIR1* promoters, we further carried out an electrophoretic mobility shift assay (EMSA). The mobility of the probes, CLE-RS2-1, CLE-RS2-3, CLE-RS2-4, and LjNIR1-3, specifically shifted when the samples were incubated with NRSYM1(531–976)-myc. In contrast, use of mutated probes, CLE-RS2-1m and LjNIR1-3m1, did not shift the mobility (Fig. 6d; Supplementary Fig. 8b; Supplementary Table 2). The band shift disappeared by the addition of competitor probes, CLE-RS2-1 or LjNIR1-3. On the other hand, the addition of a mutant competitor probe, CLE-RS2-1m, did not affect the band shift (Fig. 6e). The EMSA results showed that NRSYM1 specifically binds to NRE/NBS in the *CLE-RS2* promoter regions 1, 3, 4 and in *LjNIR1* promoter region 3. Addition of competitor CLE-RS2-3 probe inhibited the interaction between NRSYM1 and CLE-RS2-1, but the strength of competition was weaker than that of the competitor CLE-RS2-1 probe. Addition of the competitor CLE-RS2-4 probe hardly inhibited NRSYM1–CLE-RS2-1 interaction (Fig. 6e). These results suggest that NRSYM1-binding to region 1 has the strongest affinity among the three regions of the *CLE-RS2* promoter that interact with NRSYM1.

Both NRSYM1 and NIN can directly bind to the *CLE-RS2* promoter[38]. Identical amounts of NRSYM1(531–976)-myc and NIN(520–878)-myc proteins were incubated with the probe CLE-RS2-1 (Supplementary Fig. 8c). Regarding NIN-CLE-RS2-1 interaction, a shifted band was observed only when we used the greatest amount of NIN protein. In contrast, use of smaller amounts of NRSYM1 proteins caused a shift. Hence, NRSYM1 has a higher affinity for the region than NIN.

To investigate whether the identified NRE/NBS from the *CLE-RS2* and *LjNIR1* promoters are involved in NRSYM1-mediated transcriptional activation, several promoter-GUS constructs (Supplementary Fig. 8d) were introduced into the roots of WT or *nrsym1-1*, and GUS activities were quantified by real-time RT-PCR (Fig. 6f). *GUS* expression levels in *nrsym1-1* mutant roots carrying CLE-RS2-1 and LjNIR1 promoter fragments were lower than those of WT. In addition, *GUS* expression in WT was significantly reduced when the NRE/NBSs were mutated. Taken together, these results indicate that NRSYM1 regulates *CLE-RS2* and *LjNIR1* expression through direct binding to their promoters.

**Nitrate-induced control of nodule number requires CLE-RS2.** In the AON in *L. japonicus*, the CLE-RS1 and -RS2 peptides act as putative root-derived signals through interaction with their receptor, HAR1, in the shoot and transmit secondary signals that negatively regulate nodulation[13]. To elucidate the precise functions of CLE-RS1 and -RS2, we tried to determine the loss-of-function effects of *CLE-RS1* or -*RS2* genes. The CRISPR/Cas9 genome-editing system enabled us to obtain stable transgenic plants with nucleotide deletions or insertions in both genes (Supplementary Fig. 9a). We designed each gRNA to target the nucleotide sequence encoding amino acids constituting a CLE

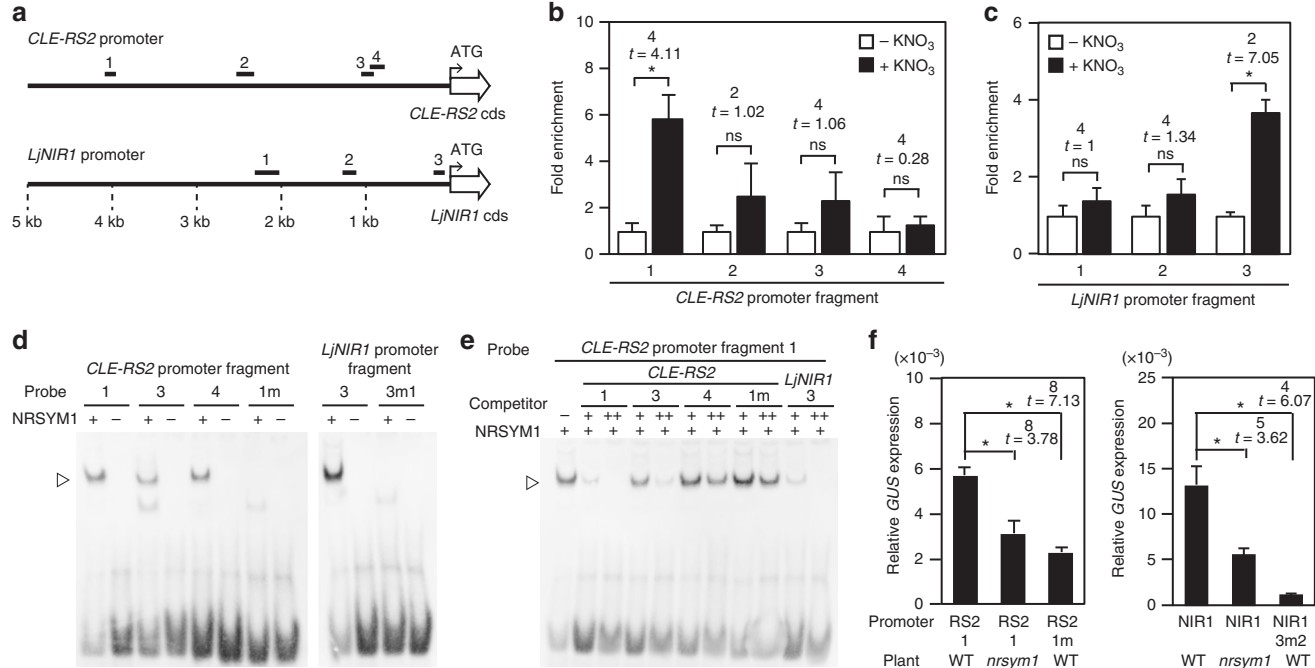

**Fig. 6** Interaction of NRSYM1 with *CLE-RS2* and *LjNIR1* promoters. **a** A schematic diagram of the location of DNA fragments used for ChIP-qPCR analyses and electrophoretic mobility shift assay (EMSA) in the 5 kb promoter regions of *CLE-RS2* and *LjNIR1*. The numbering of fragments within each promoter region corresponds to those referenced in (**b**–**e**). (**b**, **c**) qPCR analysis to examine NRSYM1 binding with the (**b**) *CLE-RS2* and (**c**) *LjNIR1* promoter regions after ChIP. DNA fragments were co-immunoprecipitated with polyclonal anti-myc antibody from chromatin suspensions prepared from *pLjUBQ:NRSYM1-myc* roots that were incubated with 0 or 10 mM $KNO_3$ for 24 h without rhizobia. The PCR products were quantified by comparison with products amplified using primers specific to *LjUBQ*. The fold enrichment of nitrate-induced NRSYM1 binding was calculated as the ratio between $+KNO_3$ and $-KNO_3$-immunoprecipitated amplification signals. Error bars indicate SEM ($n = 3$ independent pools of roots). (**d**, **e**) EMSA showing NRSYM1-binding with the NRE/NBS of the *CLE-RS2* and *LjNIR1* promoter (Supplementary Fig. 8b). Biotin-labeled probes were incubated with NRSYM1(531–976)-myc (+) or in vitro translation products without template (−). The *CLE-RS2* promoter 1m and *LjNIR1* promoter 3m1 contain mutations in the NRE/NBS of *CLE-RS2* promoter 1 and *LjNIR1* promoter 3, respectively (Supplementary Fig. 8b; Supplementary Table 2). **e** NRSYM1 (531–976)-myc and a labeled *CLE-RS2* promoter 1 probe were incubated with their respective competitors. Non-labeled probes were used as competitor DNA at an excess molar ratio (−, 1:0; +, 1:20; ++, 1:100). Arrowheads indicate locations of the band shift. **f** Real-time RT-PCR analysis of *GUS* expression in WT and *nrsym1-1* transgenic hairy roots expressing each GUS construct (Supplementary Fig. 8d). Each cDNA sample was prepared from total RNA derived from an uninoculated whorl of roots grown in the presence of 10 mM $KNO_3$ for 24 h. Transgenic roots were identified by GFP fluorescence. The expression of *GFP* was used as the reference. Error bars indicate SEM ($n = 5$ independent pools of roots). *$P < 0.05$ by Student's *t*-test. ns, not significant. Degrees of freedom are shown above the *t*-values

domain. Each gRNA that targeted *CLE-RS1* or *-RS2* had the possibility to additionally target *LjCLE49* and *CLE-RS3* among the *LjCLE* genes[40], although their off-target scores were quite low. We sequenced the *LjCLE49* and *CLE-RS3* genes in the *cle-rs1* #16, *cle-rs2* #2, and *cle-rs2* #5 lines that were used in this study and confirmed that the two *CLE* genes were unaffected in these plants. Each T0 generation of the CRISPR lines already had homozygous indel mutations (Supplementary Fig. 9a), and no other mutations in *CLE-RS1* or *-RS2* were detected. Thus, there were no chimeric mutations in the three lines. Under nitrate-free conditions, *cle-rs1* and *-rs2* plants had a normal nodulation phenotype (Fig. 7a, b), and nodule number was significantly increased in the *cle-rs1 -rs2* double-mutant (Fig. 7a), indicating that CLE-RS1 and -RS2 redundantly regulate nodule number. The nodule number for the *cle-rs1 -rs2* double mutants was reduced by introducing a 7.4-kb genomic fragment encompassing the entire *CLE-RS1* locus (Supplementary Fig. 9b). In the presence of 10 mM nitrate, nodulation was inhibited in *cle-rs1* plants to the same level as that in the WT (Fig. 7a, b). In contrast, nitrate-induced reduction in nodule number was not observed in the *cle-rs2* plants (Fig. 7a), but their nodule size was reduced by nitrate (Fig. 7b). This result led us to conclude that CLE-RS2 is involved in the nitrate-induced control of nodule number but not nodule size.

**NRSYM1 accumulates in the nucleus in response to nitrate**. Although we have shown that NRSYM1 can directly regulate nitrate-inducible gene expression (Fig. 6), *NRSYM1* expression per se was not affected by nitrate (Supplementary Fig. 6a). The subcellular localization of AtNLP6 and AtNLP7 proteins that belong to the same clade as NRSYM1 is regulated by nitrate; nuclear localization of the proteins is retained in the presence of nitrate[41,42]. We, thus, examined the subcellular localization of NRSYM1 by immunohistochemistry. The subcellular localization of NRSYM1 was indirectly determined using transgenic roots carrying the *pLjUBQ:NRSYM1-myc* construct combined with detection with polyclonal anti-myc primary antibodies and secondary antibodies containing Alexa Fluor 488. Although NRSYM1 was barely detected in nuclei under nitrate-free conditions (Fig. 8a, f), the protein was predominantly localized in nuclei within 20 min of nitrate treatment (Fig. 8b, f). Nuclear localization was also observed after 24 h of nitrate treatment (Fig. 8c, f). Moreover, nuclear accumulation of NRSYM1 was reversible when nitrate was removed (Fig. 8d, f). The addition of leptomycin B (LMB), an inhibitor of nuclear export[43], inhibited the export of NRSYM1 from nuclei when nitrate-supplied roots were moved to a N-free medium (Fig. 8e, f). On the basis of these

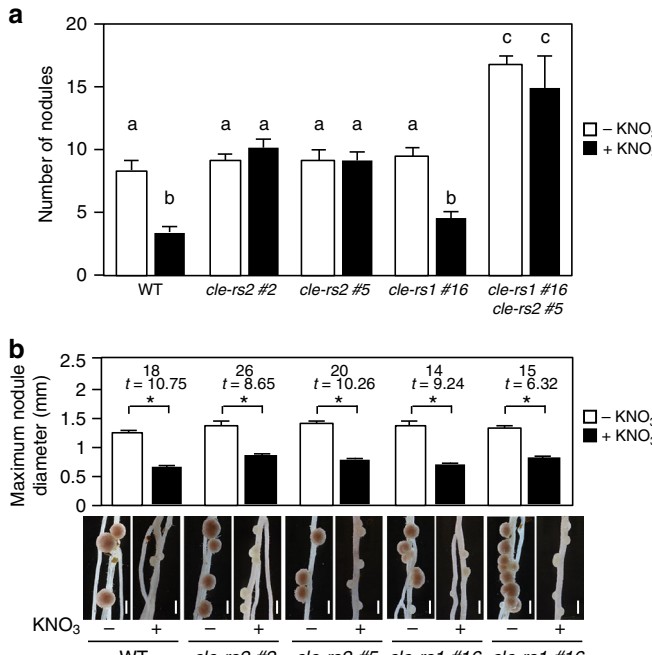

**Fig. 7** Effects of loss-of-function mutations in *CLE-RS1/2* on nodulation. **a** The number of nodules and (**b**) maximum nodule diameter (mm) in WT, the *cle-rs1* mutants (T2), the *cle-rs2* mutants (T2), and the *cle-rs1 -rs2* double mutants grown in the presence of 0 or 10 mM KNO$_3$ ($n = 7$–14 plants). The *cle-rs1 -rs2* double mutants were obtained by crossing the single mutants. The *cle-rs1* and *-rs2* mutants were obtained by the CRISPR/Cas9 genome-editing system (Supplementary Fig. 9a). Error bars indicate SEM. Columns with the same lower-case letter indicate no significant difference (Tukey's test, $P < 0.05$; **a**). *$P < 0.05$ by Student's *t*-test. Degrees of freedom are shown above the *t*-values (**b**). Scale bars: 1 mm

data we propose that nuclear localization of NRSYM1 is regulated by nitrate as is the case for AtNLP6/7.

## Discussion

To balance the gains and costs associated with root nodule symbiosis, plants control the number of nodules per root system using the AON system. In addition, plants can stop the symbiosis and save the cost if sufficient nitrogen is available in their surrounding environment. In this study, we identified NRSYM1 that acts as a key regulator in the latter mechanism. Under nitrate-sufficient conditions, unstoppable nodulation in *nrsym1* mutants diminished shoot growth, demonstrating the significance of the NRSYM1-mediated control of root nodule symbiosis for plant growth. More than 30 years ago, a similar approach to this screening study isolated several soybean mutants that all had not only nitrate-tolerant phenotypes but also formed an excess number of nodules[24]. Indeed, it turned out that the gene responsible for some of the mutants encodes an LRR-RLK that acts as a pivotal factor in the AON[18]. Since then the AON has been proposed to have another role in mediating the nitrate-induced control of root nodule symbiosis. Unlike the canonical AON mutants, the *nrsym1* mutants form a normal number of nodules and show tolerance to all examined nitrate-affected processes. Genetic analysis of *NRSYM1*, *HAR1*, and *CLE-RS2* indicated that the three genes act in the same genetic pathway. In addition, we have shown that NRSYM1 directly regulates *CLE-RS2* expression in response to nitrate, providing a direct molecular link between nitrate and nodulation. After the activation of

*CLE-RS2*, the downstream signaling pathway may be identical to that for AON, where the produced CLE-RS2 peptide functions as a root-derived signal through the interaction with the shoot-acting HAR1. In nitrate-sufficient conditions, the *cle-rs2* or *har1* mutants showed tolerance to nitrate-induced control of nodule number but not to other nitrate-affected processes. Hence, we hypothesize that the nitrate-induced NRSYM1>CLE-RS2>HAR1 signaling pathway plays a role predominantly in the control of nodule number (Fig. 9a). In contrast, NRSYM1 is likely to use different downstream targets to achieve AON-independent regulation of other processes such as rhizobial infection, nodule growth, and nitrogen fixation activity (Fig. 9a). This notion is consistent with a previous suggestion that there are both an AON-dependent and an -independent mechanism in nitrate-induced control of root nodule symbiosis[7,25,26]. In most experiments of this study, we used 10 mM KNO$_3$, a nitrate concentration sufficient to inhibit symbiosis. The application of much lower KNO$_3$ concentrations (for example, 200 μM) induced *CLE-RS2* expression in an NRSYM1-dependent manner. The activation of *CLE-RS2* by low concentrations of nitrate seems to be insufficient for inhibiting nodulation. Thus, there might be an unidentified mechanism in response to high nitrate concentration that plays a role in inhibiting nodulation in parallel or downstream of *CLE-RS2* activation.

The amino-acid sequence of the CLE domain is indistinguishable between CLE-RS1 and -RS2, and the two peptides have similar negative activity in the control of nodulation[7]. In this study, the creation of loss-of-function mutants enabled us to understand the conserved and diverse functions of these genes. The *cle-rs1 -rs2* double mutants have more nodules, whereas the single mutations do not affect nodule number, demonstrating that CLE-RS1 and -RS2 have redundant functions in controlling nodule number. However, the *cle-rs1 -rs2* double-mutant phenotype is milder than that shown by their receptor mutant, *har1*. In Arabidopsis, although many CLE peptides are involved in diverse developmental processes and their putative receptors are identified[44], there is only one case where the loss-of-function phenotype of a ligand is almost identical to that of its receptor[45,46], implying that loss-of-function effects in a *CLE* gene are usually masked because of functional redundancy. Therefore, there may be a higher-order functional redundancy in CLE peptides regarding their control of nodule number. CLE-RS3 is, at least, now identified as a *HAR1*-dependent negative regulator of nodulation, and there are other *LjCLE* genes whose expression is induced during nodulation[8]. Under nitrate-sufficient conditions, as mentioned above, the *cle-rs2* mutants can produce a normal number of nodules, whereas the *cle-rs1* mutants were affected by nitrate. These results indicate that CLE-RS2 has a dual role in regulating the AON and nitrate-induced control of nodule number; CLE-RS1's function is specific to the former process. The different roles of the two genes agree with the observation that the expression of *CLE-RS2* is induced by nitrate but not by *CLE-RS1*.

The designation of the NLP family originates from the symbiosis-specific transcription factor NIN[30]. NLP consists of an N-terminal conserved domain responsible for nitrate response, an RWP-RK DNA-binding domain, and a conserved PB1 domain involved in protein–protein interaction[29]. In Arabidopsis, a non-leguminous plant, NLPs are considered as master regulators of the nitrate response because AtNLP6/7 regulates many nitrate-responsive genes[41,47]. In Arabidopsis nitrate signaling, NITRATE TRANSPORTER1.1 (NRT1.1/CHL1) is thought to act as a nitrate sensor, and the functions of some *M. truncatula* proteins belonging to NRT1 family have been characterized[48–50]. More recently, the molecular link between nitrate-sensing and AtNLP7-mediated activation of nitrate-inducible genes has been elucidated, where Ca$^{2+}$-sensor protein kinases (CPKs) act as master

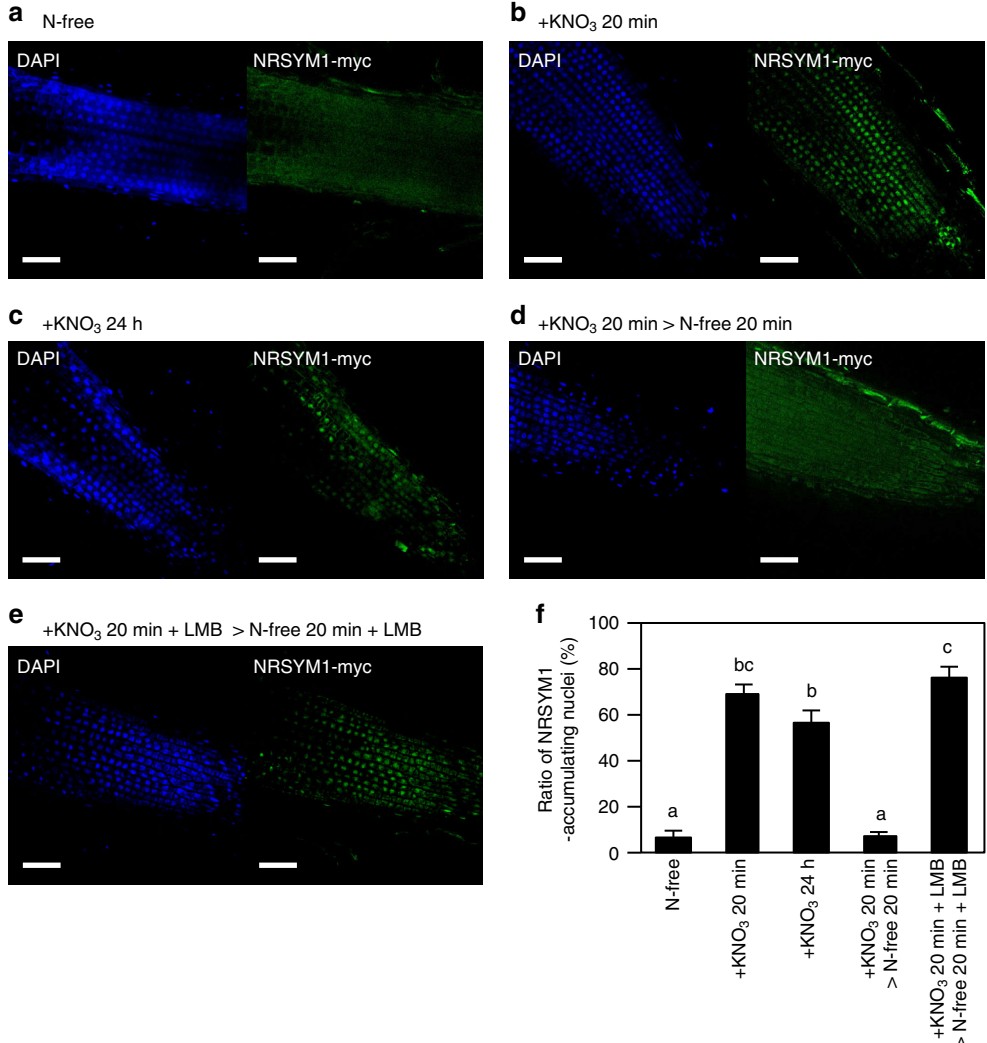

**Fig. 8** Subcellular localization of NRSYM1. **a–e** Immunohistochemistry of the NRSYM1-myc protein in root apical cells. A polyclonal anti-myc antibody and an antibody conjugated to Alexa Fluor 488 (green signal) were used as primary and secondary antibodies. Nuclei were visualized with DAPI (blue signal). Plants with transgenic hairy roots carrying the *pLjUBQ:NRSYM1-myc* construct were grown in the absence of nitrate for 3 days. N-starved plants were transferred to (**a**) N-free or (**b**) 10 mM $KNO_3$ medium for 20 min or (**c**) 24 h. **d** N-starved plants were transferred to 10 mM $KNO_3$ medium for 20 min, and then to N-free medium for 20 min. **e** N-starved plants were first incubated with leptomycin B (LMB) in a N-free medium for 3 h, and then transferred to 10 mM $KNO_3$ medium with LMB for 20 min and to N-free medium with LMB for 20 min. Scale bars: 50 μm. **f** The ratio of NRSYM1-accumulating nuclei. Using respective fluorescent images, the percentages of nuclei having green signals among all DAPI-stained nuclei were calculated. Error bars indicate SEM ($n = 5$ roots). Columns with the same lower-case letter indicate no significant difference (Tukey's test, $P < 0.05$)

regulators that orchestrate the primary nitrate response[51]. *L. japonicus* has five NLPs and NIN; one of the NLP Lj1g3v2295200 (LjNLP1) is reported to bind to NRE and to respond to nitrate[29]. The partial loss of amino-acid residues in the N-terminal-conserved domain of NIN is thought to be associated with the protein's loss of nitrate responsiveness[29]. Alternatively, NIN has a new function in playing a role as a necessary and sufficient factor for nodulation[34–37]. Therefore, it seems that the basal function of NLPs in plants can be related to nitrate response. Thus, the emergence of NIN provides an example of neofunctionalization during the evolution of legumes, where after gene duplication one of the NLPs might have been released from functional constraints, enabling it to accumulate mutations and to acquire a new function[52]. The *nrsym1* mutation caused the *Atnlp7*-like nitrate starvation phenotypes in non-symbiotic conditions[53], and NRSYM1 has a role in regulating the expression of general nitrate-responsive genes involved in nitrate assimilation. In addition, constitutive expression of *AtNLP6* or *AtNLP7* partially

rescued the *nrsym1* nodulation phenotype. These observations suggest that the original function as a regulator of a general nitrate response is maintained in NRSYM1. Furthermore, we have revealed that NRSYM1 has a crucial function relevant to root nodule symbiosis by regulating nodulation-specific genes such as *CLE-RS2* (Fig. 9b). Therefore, in terms of the evolution of the NLP family in legumes, NLPs might have evolved toward opposite directions; that is, NRSYM1 and NIN have acquired negative and positive roles, respectively, in the control of root nodule symbiosis. Given that NRSYM1 and NIN share *CLE-RS2* as a direct target gene, the NRSYM1>CLE-RS2 transcriptional regulatory module might be a prototype for AON. In ChIP experiments, we did not detect NRSYM1 binding to region 3 or 4 of the *CLE-RS2* promoter, to which NIN binds[38]. In contrast, the EMSA results suggest that NRSYM1 can bind to regions 3 and 4 in addition to region 1. The difference in these results may be related to differences between in vivo and in vitro experiments. The EMSA competition assay showed that NRSYM1 had the

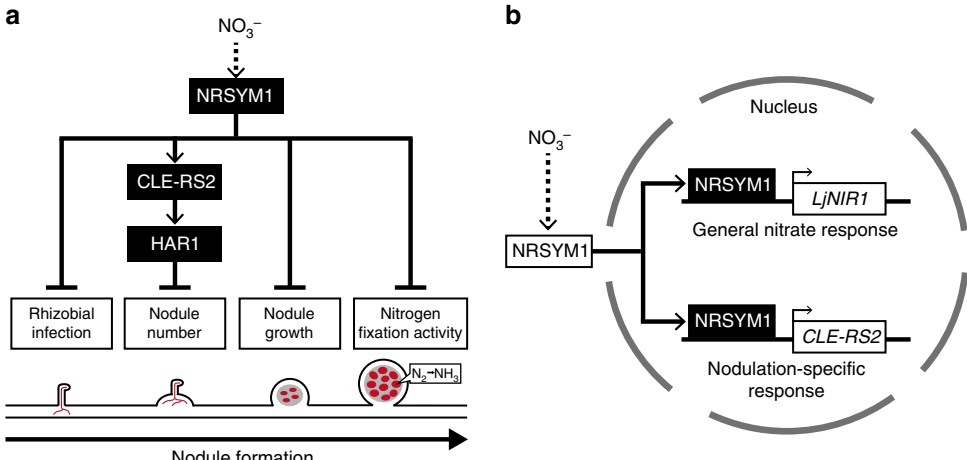

**Fig. 9** Model for the control of root nodule symbiosis in response to nitrate. **a** Sequential progress of nodulation is shown. In response to nitrate, NRSYM1 regulates pleiotropic phases of root nodule symbiosis, including rhizobial infection, nodule number, nodule growth, and nitrogen fixation activity. Whereas NRSYM1 activates the CLE-RS2>HAR1 signaling pathway leading to the negative regulation of nodule number, NRSYM1 is likely to use different downstream targets to achieve the regulation of other nitrate-affected processes. Red lines and red cells, respectively, indicate the infection threads and rhizobia-colonized cells. **b** A model for cellular-level NRSYM1 function. Nuclear localization of NRSYM1 is controlled by nitrate. In the nucleus, NRSYM1 regulates both the general nitrate response and root nodule symbiosis by directly regulating related genes such as *LjNIR1* and *CLE-RS2*

strongest affinity for region 1 with lesser affinities for regions 3 and 4. In addition, the promoter-GUS assay result suggests that interaction between NRSYM1 and region 1 may be sufficient to activate *CLE-RS2*.

Like Arabidopsis NLP6/7, the expression of *NRSYM1* is not induced by nitrate, suggesting that post-translational regulation of NRSYM1 provides it with function[31]. As nuclear localization of NRSYM1 is regulated by nitrate as well as AtNLP6/7[41,42] (Fig. 9b), the nitrate-induced nuclear localization may be a feature of the NRSYM1/AtNLP6/7 clade of the NLP family. Recently, CPK-dependent phosphorylation of AtNLP7 was reported to regulate its nuclear retention[51]. AtNLP8, a master regulator of nitrate-promoted seed germination, was shown to be localized in nuclei independently of nitrate; this finding suggests that an unknown mechanism provides AtNLP8 with transcriptional regulatory activity[54]. Nitrate-induced control of root nodule symbiosis is known to be a reversible process dependent on nitrate availability[55]. Future investigations of the detailed mechanism for reversible NRSYM1 nuclear localization might provide great insight into the underlying mechanism of the phenomenon.

## Methods

**Plant materials and growth conditions**. The Miyakojima MG-20 ecotype of *L. japonicus*[56] was used as WT in this study. The Gifu B-129 ecotype of *L. japonicus*[57] was used as a crossing partner for map-based cloning of *NRSYM1*. The *nrsym1-1* and *nrsym1-2* mutants were isolated from the M₂ generation of WT that had been mutagenized with 0.4% EMS. M₂ seeds were collected from ~10,000 M₁ plants provided from LegumeBase. A description of the *har1-7* mutants was published previously[9]. Plants were grown with or without *Mesorhizobium loti* MAFF 303099 in autoclaved vermiculite with Broughton and Dilworth (B&D) solution[58] that does not contain a nitrogen source. The plants were grown under a 16 h light/8 h dark cycle at 24 °C in a growth cabinet. For measurement of relative nodule size, plants were grown with or without *M. loti* on a 1% agar plate containing B&D medium under the same light conditions. For the nitrate response assay, different concentrations of $KNO_3$ (0–50 mM) were supplemented into B&D solution. The solution was exchanged every 7 days after inoculation with newly prepared B&D solution containing different concentrations of $KNO_3$ to maintain the original $KNO_3$ concentrations.

**Acetylene reduction assay**. The nitrogenase activity of nodules was indirectly determined by measuring the acetylene reductase activity[59]. Nodulated roots detached from intact plants were placed in 20 ml vials, and after injection of acetylene they were incubated for 30 min at 25 °C. Then, the amount of ethylene produced was measured using a gas chromatography.

**Genome-resequencing of the *nrsym1* mutants**. The leaves of the *nrsym1* mutants were ground in liquid nitrogen using a mortar and pestle. Genomic DNA was isolated using a DNeasy Plant Mini Kit (Qiagen). The quality of purified genomic DNA was evaluated by a Quant-iT dsDNA BR Assay Kit (Invitrogen). For whole-genome shotgun sequencing of the *nrsym1* mutants, a library was constructed using a TruSeq DNA Sample Prep kit (Illumina) following the manufacturer's instructions. The quality of the library was checked using an Agilent 2100 Bioanalyzer and quantified using a KAPA Library Quant Kit (Kapa Biosystems). Paired-end 101 bp × 2 sequencing was performed using an Illumina HiSeq 2000 instrument (Illumina). Short reads were mapped against the *L. japonicus* genome assembly build 2.5[60] by Bowtie2[61]. The resulting data in the SAM format were converted to a binary equivalent BAM format and sorted using the samtools software package[62]. Variant calling was performed with samtools and bcftools.

**Constructs and hairy root transformation of *L. japonicus***. The primers used for PCR are listed in Supplementary Table 3. For the complementation analysis of the *nrsym1* mutants, a 7.5-kb genomic DNA fragment including the *NRSYM1* candidate gene was amplified by PCR from WT genomic DNA. This fragment, including a 2.3-kb sequence directly upstream of the initiation codon, was cloned into pCAMBIA1300-GFP-LjLTI6b[63]. The coding sequences (cds) of *NRSYM1* or *AtNLP6/7* were, respectively, amplified by PCR from template cDNAs prepared from WT *L. japonicus* or Arabidopsis Col-0 plants and were cloned into the pENTR/D-TOPO vector (Invitrogen). The insert was transferred into pUB-GW-GFP[64] by the LR recombination reaction. For the *NRSYM1* expression analysis, a 2.3-kb fragment of the *NRSYM1* promoter region was amplified by PCR from WT genomic DNA and cloned upstream of the *GUS* gene in the pCAMBIA1300-GUS-GFP-LjLTI6b vector[8]. For ChIP and immunohistochemistry analysis, the *NRSYM1* cds without a stop codon was amplified by PCR from template cDNA prepared from WT roots and cloned into the pENTR/D-TOPO vector. The insert was transferred into pGWB20[65] by the LR recombination reaction in order to express a C-terminal fusion to a 10xMyc (myc) tag. Using the resulting construct as a template, the *NRSYM1-myc* fragments were amplified by PCR and cloned into the pENTR/D-TOPO vector. The insert was transferred into pUB-GW-GFP by the LR recombination reaction. To make the construct for in vitro translation of NRSYM1, a part of the *NRSYM1* cds (1591-2931) was amplified by PCR from template cDNA prepared from WT. The fragment was replaced with NIN that had been previously cloned into the downstream region of a 3xMyc tag in the pENTR1A vector (Invitrogen)[36]. The resulting *NRSYM1* (1591-2931)-*myc* fragments were amplified by PCR and cloned into the pF3K-WG (BYDV) Flexi vector (Promega). The NIN-myc construct for in vitro translation was described previously[36]. For promoter-GUS analysis, a 2.7-kb fragment of the *LjNIR1* promoter region (pLjNIR1) was amplified by PCR from WT genomic DNA. To make the *LjNIR1* promoter region lacking NRE (pLjNIR1-3m2), genomic fragments of the regions upstream and downstream of NRE were, respectively, amplified by PCR from WT genomic DNA. Each fragment was inserted upstream of the *GUS* gene in the pCAMBIA1300-GUS-GFP-LjLTI6b vector. The *pCLE-RS2-1:GUS* and *pCLE-RS2-1m:GUS* constructs are identical to CLE-RS2 region 1 and CLE-RS2 region 1 S1m as described previously[38]. For the complementation analysis of the *cle-rs1 -rs2* double mutants, a 7.4-kb genomic DNA fragment including the *CLE-RS1* gene was amplified by PCR from WT genomic DNA. This fragment, including a 5.9-kb sequence directly

upstream of the initiation codon, was cloned into pCAMBIA1300-GFP. The recombinant plasmids were introduced into *A. rhizogenes* and were transformed into roots of *L. japonicus* plants by a hairy root transformation. Seeds were germinated on germination medium (1/2x Gamborg's B5 salt mixture (Wako), 1/2x Gamborg's vitamin solution (Sigma), 1% sucrose, 1% agar) in a growth cabinet (24 °C dark for first 2 days, 24 °C 16 h light/8 h dark cycle for next 2 days). *A. rhizogenes* AR1193 strains harboring each construct were streaked on YEP plate with appropriate antibiotics for 2 days at 28 °C. Seedlings were placed in the *A. rhizogenes* suspension and then cut at the base of the hypocotyls. The seedlings with cotyledons were transferred onto hairy root medium (1× Gamborg's B5 salt mixture, 1× Gamborg's vitamin solution, 2% sucrose, 1% agar) and were grown in a growth cabinet (24 °C dark for first 1 day, 24 °C 16 h light/8 h dark cycle for next 2 days). Then, the plants were transferred onto fresh hairy root medium containing 12.5 μg ml⁻¹ meropen and were grown for 7–10 days in a growth cabinet (24 °C 16 h light/8 h dark cycle). Transgenic roots were identified by GFP fluorescence, and the plants with transgenic hairy roots were used for further experiments.

**Constructs and stable transformation of *L. japonicus*.** To create CRISPR/Cas9 constructs of *CLE-RS1* or *-RS2*, targeting sites in the genes were designed using the CRISPR-P program (http://cbi.hzau.edu.cn/crispr/)[66]. Oligonucleotide pairs (Supplementary Table 3) were annealed and cloned into a single guide RNA (sgRNA) cloning vector, pUC19_AtU6oligo, as previously described[67]. Then, the sgRNA expression cassette prepared in pUC19_AtU6oligo was excised and replaced with OsU3:gYSA in pZH_gYSA_FFCas9, an all-in-one binary vector harboring a sgRNA, Cas9, and an HPT expression construct, as previously described[67]. The recombinant plasmids were introduced into *L. japonicus* plants by *A. tumefaciens*-mediated stable transformation. Seeds were germinated on germination medium described above in a growth cabinet (24 °C dark for first 2 days, 24 °C 16 h light/8 h dark cycle for next 2 days). *A. tumefaciens* AGL1 strains harboring each construct were streaked on YEP plate with appropriate antibiotics for 2 days at 28 °C. Seedlings were placed in the *A. tumefaciens* suspension and then their hypocotyls were cut into about 3 mm pieces. The hypocotyl pieces were placed onto the top of pilled filter papers saturated with co-cultivation medium (1/10× Gamborg's B5 salt mixture, 1/10× Gamborg's vitamin solution, 0.2 μg ml⁻¹ BAP, 0.05 μg ml⁻¹ NAA, 5mM MES (pH 5.2), 20 μg ml⁻¹ acetosyringone, pH 5.5) and were incubated in a growth cabinet (21 °C dark) for 6 days. After that, the hypocotyl pieces were transferred to callus induction medium (1× Gamborg's B5 salt mixture, 1× Gamborg's vitamin solution, 2% sucrose, 0.2 μg ml⁻¹ BAP, 0.05 μg ml⁻¹ NAA, 10 mM (NH₄)₂SO₄, 0.3% phytagel, 12.5 μg ml⁻¹ meropen, 15 μg ml⁻¹ Hygromycin B, pH 5.5) and were incubated in a growth cabinet (24 °C 16 h light/8 h dark cycle) for 2–3 weeks. The hypocotyl pieces were transferred to fresh callus induction medium every 5 days. When calluses became more than 1 mm in size, they were detached from the hypocotyls and transferred onto callus medium without hygromycin B, and were incubated for 3–7 weeks in a growth cabinet (24 °C 16 h light/8 h dark cycle) until leaf primordia became visible. The calluses were transferred onto new medium every 5 days. The calluses with leaf primordia then were transferred to shoot elongation medium (1× Gamborg's B5 salt mixture, 1× Gamborg's vitamin solution, 2% sucrose, 0.2 μg ml⁻¹ BAP, 0.3% phytagel, 12.5 μg ml⁻¹ meropen, pH 5.5), and incubated until their shoot length became ~1 cm. Individual shoots were detached from calluses and transferred to root induction medium (1/2× Gamborg's B5 salt mixture, 1/2× Gamborg's vitamin solution, 1% sucrose, 0.5 μg ml⁻¹ NAA, 0.4% phytagel, 12.5 μg ml⁻¹ meropen, pH 5.5), and incubated for 10 days. Then, they were transferred to root induction medium without NAA and cultivated until their root length became ~2–3 cm. Thereafter, the transgenic plants were transplanted into vermiculite for further cultivation.

**Expression analysis.** The primers used for PCR are listed in Supplementary Table 3. Total RNA was isolated from respective organs using the PureLink Plant RNA Reagent (Invitrogen). First-strand cDNA was prepared using the ReverTra Ace qPCR RT Master Mix with gDNA Remover (Toyobo). Real-time RT-PCR was performed using a Light Cycler 96 System (Roche) or a 7900HT Real-Time PCR system (Applied Biosystems) with a THUNDERBIRD SYBR qPCR Mix (Toyobo) according to the manufacturer's protocol. The expression of *LjUBQ* was used as the reference.

**Grafting.** Seeds were germinated on 1% agar plates in a growth cabinet (24 °C dark for first 2 days, 24 °C 16 h light/8 h dark cycle for next 2 days). First, seedlings were cut perpendicularly at the hypocotyls with a scalpel blade. A shoot scion was then sliced at an angle and inserted into a short vertical slit (~2 mm) made on a rootstock. Grafted plants were sandwiched by two filter papers saturated with sterilized water in plastic plates for 3 days, and transferred to sterilized vermiculite pots.

**ChIP-qPCR assay.** Chromatin suspensions were prepared from 1 g of hairy roots constitutively expressing *NRSYM1-myc* with or without 10 mM KNO₃. Roots were fixed with 1% formaldehyde in MC buffer (10 mM potassium phosphate (pH 7.0), 50 mM NaCl, 0.1 M sucrose) for 10 min under vacuum. The reaction was stopped by adding 0.125 M glycine, and the roots were washed three times with MC buffer. The fixed roots were powdered with a mortar and pestle in liquid nitrogen,

suspended with 15 ml of M1 buffer (10 mM potassium phosphate (pH 7.0), 0.1 M NaCl, 10 mM β-mercaptoethanol, 1 M 2-methyl 2,4-pentane-diol) supplemented with 1 mM PMSF and Complete Protease Inhibitor Cocktail (Roche Diagnostics). The crude extract was filtered through two layers of Miracloth and washed with 15 ml of M1 buffer. The filtrate was centrifuged at 1600×g for 15 min at 4 °C. The pellet was washed four times, each with 1 ml of M2 buffer (M1 buffer with 10 mM MgCl₂ and 0.5% Triton X-100), and once with M3 buffer (M1 buffer without 2-methyl 2,4-pentanediol). After centrifugation at 1600×g for 15 min, the pellet was resuspended in 1 ml of Sonication buffer (10 mM potassium phosphate (pH 7.0), 0.1 mM NaCl, 0.5% Sarkosyl, 10 mM EDTA) that was supplemented with 1 mM PMSF and Protease Inhibitor Cocktail, incubated for 20 min on ice, and sonicated six times for 10 s at 40 % power output using a Sonics Vibra-Cell vcx 130. The chromatin suspension was centrifuged twice at 14,000×g for 15 min at 4 °C. An equal volume of immunoprecipitation (IP) buffer (50 mM Hepes (pH 7.5), 150 mM KCl, 5 mM MgCl₂, 1% Triton X-100, 0.05% SDS) was added to the supernatant. Then, 2 μg of anti-myc polyclonal antibodies (Santa Cruz Biotechnology Inc) were added to the remaining suspension and the mixture was incubated for 6 h at 4 °C. After centrifugation, 40 μl of protein A/G agarose (25% slurry; Santa Cruz Biotechnology Inc) was added to the supernatant, and this mixture was rotated for 2 h at 4 °C. After washing five times with IP buffer, the immunoprecipitate was eluted twice with 200 μl of elution buffer (0.1 M glycine, 0.5 M NaCl, 0.05% Tween-20, pH 2.8), and then 100 μl of 1 M Tris buffer (pH 9.0) was added into the combined eluates. DNA sample was treated with 10 mg ml⁻¹ RNase A for 30 min at 37 °C, followed by 18.2 mg ml⁻¹ proteinase K for 30 min at 37 °C. After incubation for 6 h at 65 °C, a second aliquot of proteinase K was added and incubated at 37 °C for 1 h to reverse the formaldehyde crosslinks. After cooling, proteins were extracted by phenol–chloroform extraction. After ethanol precipitation, the DNA was dissolved in 30 μl of 10 mM Tris-HCl (pH 8.0). qPCR was performed using a Light Cycler 96 System (Roche) with a KOD SYBR qPCR Mix (Toyobo) according to the manufacturer's protocol. The primers used for PCR are listed in Supplementary Table 3.

**EMSA.** The NRSYM1 (531–976)-myc and NIN (520–878)-myc proteins were synthesized using the TNT SP6 high-yield wheat germ protein expression system (Promega) according to the manufacturer's protocol. Western blotting was performed with an adjustment for loading equal amounts of the proteins. The in vitro translation products were separated in 10% polyacrylamide gels with running buffer (25 mM Tris, 192 mM glycine, 0.1% SDS). Proteins in the gels were transferred to Immobilon-P membrane (Millipore) in transfer buffer (25 mM Tris, 192 mM glycine, 20% methanol). Membranes were blocked overnight at room temperature in 5% skim milk and washed in TBST buffer (50 mM Tris-HCl (pH 7.5), 0.15 M NaCl, 0.1% Tween-20). Membranes were incubated with 0.5 μg ml⁻¹ anti-myc polyclonal antibody (Santa Cruz Biotechnology Inc) in TBST for 2 h at 28 °C. Membranes were washed three times in TBST and incubated for 1 h with a 1:5000 dilution of goat anti-rabbit IgG antibodies (Amersham Pharmacia). EMSA was performed as follows. The in vitro translation products produced without templates were used as the control. For preparing probes and competitors, single-strand oligonucleotides were annealed to form dsDNA. Probes were labeled with biotin. The in vitro translation products containing NRSYM1 (531–976)-myc or NIN (520–878)-myc were incubated with probes in 10 μl of EMSA DNA-binding buffer (20 mM HEPES-KOH (pH 7.9), 50 mM KCl, 1 mM dithiothreitol , 4% glycerol, 0.1% Triton X-100) supplemented with 2 mg bovine serum albumin (BSA) and 1 mg Salmon Sperm DNA, for 30 min at room temperature. Reactants were separated in 15% polyacrylamide gels with 0.5× TBE buffer. DNA in the gels were transferred to Hybond-N+ membrane (Amersham Pharmacia) in 0.5× TBE buffer. Membranes were then incubated with a 1:1000 dilution of streptavidin–horseradish peroxidase conjugate (Thermo) in Nucleic Acid Detection Blocking Buffer (Thermo) for 1 h at 28 °C. Signals were detected using the ECL Prime Western Blotting Detection Reagent (Amersham Pharmacia) and LAS-4000mini (Fujifilm). Oligonucleotides used for probe synthesis are listed in Supplementary Table 2.

**Immunohistochemistry.** Plants with transgenic hairy roots carrying the *pLjUBQ: NRSYM1-myc* construct were nitrogen-starved by growing them in the absence of nitrate for 3 days, followed by the addition of 10 mM KNO₃. LMB (100 nM; Sigma) was used for the nuclear export inhibition experiment. The roots were fixed in 3% paraformaldehyde in MTSB buffer (50 mM PIPES-KOH (pH 7.0), 5 mM EGTA, 5 mM MgSO₄) for 40 min under vacuum. The fixed roots were then incubated for 15 min in 0.1 M glycine, 0.1 M NH₄Cl in MTSB buffer, and washed five times with MTSB buffer. After washing six times for 10 min with wash buffer (MTSB buffer with 0.1% Triton X-100), roots were pre-incubated in 3% BSA in wash buffer at room temperature for 1 h and then incubated with a 1:200 dilution of an anti-myc polyclonal antibody (Santa Cruz Biotechnology Inc) in 3% BSA in wash buffer at room temperature overnight. After washing eight times for 10 min with wash buffer, the roots were incubated with a 1:500 dilution of anti-sheep IgG-Alexa fluor 488 (Invitrogen) in 3% BSA in wash buffer at room temperature for 3 h. Finally, the roots were washed eight times for 12 min with wash buffer and five times for 10 min with MTSB buffer. Before observing the signal, the roots were stained with 5 μg ml⁻¹ 4', 6-diamidino-2-phenylindole (DAPI, Dojindo) for 15 min. Fluorescent images were obtained using a LSM700 confocal laser-scanning microscope (Carl Zeiss) equipped with ZEN (Carl Zeiss). The obtained images were analyzed using Image J; first, the threshold of the green signals derived from

NRSYM1-myc was set equally among the images, and then the ratio of the number of the nuclei with green signals was quantified against the number of the total, namely DAPI-stained, nuclei in every image.

**Data Availability**. Sequence data from this article can be found in the GenBank/EMBL data libraries under the following accession number: NRSYM1, LC230020. Data from the short reads from the WT (MG-20), *nrsym1-1*, and *nrsym1-2* genomic DNA were deposited in the DNA Data Bank of Japan Sequence Read Archive under the accession number DRA005940.

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

## Acknowledgements

We thank LegumeBase for providing EMS-treated M$_2$ seeds; Dr Friedrich Fauser (Carnegie Institution for Science), Dr Simon Schiml, and Professor Holger Puchta (University of Karlsruhe) for providing the *Cas9* gene; Dr Masaki Endo, Dr Seiichi Toki (National Food Research Institute, NARO), Mr Masafumi Mikami (Yokohama City University), and Dr Tsuyoshi Nakagawa (Shimane University) for providing plasmids; Dr Jens Stougaard (Aarhus University) for providing *A. rhizogenes* AR1193 strain; Dr Takuji Ohyama (Tokyo University of Agriculture) for sharing a reference. This work was supported by the Cooperative Research Grant of the Plant Transgenic Design Initiative by Gene Research Center, University of Tsukuba, National Institute for Basic Biology (NIBB), Model Plant Research Facility, NIBB BioResource Center, Functional Genomics Facility, NIBB Core Research Facilities, the Japan Advanced Science Research Network, MEXT/JSPS KAKENHI, Japan (16H01457 to T.Su., 15H05962 to S.M., and 25291066 and 22128006 to M.K.), by JST ERATO, Japan (JPMJER1502 to S.B. and T.Su.), and Grant-in-Aid for JSPS Research Fellows (17J02948 to H.N.).

## Author contributions

H.N., M.K. and T.Su. designed the experiments. H.N., S.T., Y.H., M.I., Y.S., S.M., S.B., K. M., T.So. and T.Su. performed experiments and analyzed the data. H.N. and T.Su. wrote the paper.

## Additional information

**Competing interests:** The authors declare no competing financial interests.

