## [Peer Review File · Nature Communications]

Reviewers' comments:

Reviewer #1 (Remarks to the Author):

Nishida et al used a genetic screen to identify a new gene, NRSYM1, mediating nitrate control of nodulation and molecular genetics to characterize the function of the NRSYM1 as a transcriptional regulator. Careful phenotypic analysis involving double mutant analyses and grafting then allowed the authors to distinguish the impact of nitrate on autoregulation of nodule numbers from an independent role in regulation of developmental and functional processes essential for symbiosis. Both of these pathways turned out to be controlled by NRSYM1 but via independent mechanisms. The role of nitrate regulation of nodulation has for many years been a puzzle and the identification of NRSYM1 as central hub is a major achievement. The results presented are novel and interesting and the conclusions are generally well supported by experimental evidence. The research presented is of general interest for the community working on plant-microbe interactions, the community working on nitrogen regulation of plant growth and the identification of a new component influencing the long-range shoot to root communication is of interest for scientist working on multicellular organisms. Altogether this work contributes novel and significant results and improves our understanding of nitrogen regulation in developmental processes of plants and symbiosis in particular.

These are some issues in the experimental work and in the presentation, that needs the authors attention. Specific comments are listed below.

Experimental work:

The genetic inheritance of the *nrsym1* mutation is not included although a map based cloning approach was used. Is the mutation recessive? There should be segregation data available from the mapping. Please also include references presenting the MG20 and Gifu accessions as model legumes and their use as mapping partners in order to inform readers outside the field.

The CRISPR/Cas generated mutations of CLE-RS1 and CLE-RS2 are not clearly described. Which region of the genes were targeted and were possible off-targets in other CLE genes considered in the design? How were the mutants characterized molecularly to ensure they were not chimeric? Was segregation of the mutations analyzed? Were the mutants complemented to make sure the observed effects were due to inactivation of the CLE-RS1 and CLE-RS2 genes and no other members of the large CLE family that may be off-targets? These questions are important for the conclusions drawn.

ChIP was used to show *in vivo* binding of NRSYM1 to region 3 of CLE-RS2 promoter in order to support the NRSYM1-CLE-RS2-HAR1 pathway. Any results for the importance of NRSYM1 binding from deletion of region 3 in the CLE-RS2 promoter?

Presentation:

The wording is not always optimal. Examples are listed below but the list is probably not complete.

The biological processes that are central for the manuscript are not well-presented in the abstract " To balance the profits and costs associated with the symbiosis, plants have developed two major strategies; plants can maintain nodule number and can stop the symbiosis upon nitrogen availability in the soil." This statement is not incorrect but the meaning is unclear, even for experts. There are many ways to rephrase the sentence and I can suggesttwo strategies for adapting to nitrogen availability in the soil; plants can regulate nodule numbers and/or stop development or function of processes central for symbiosis. The point is to be precise so readers outside the field understand.

"Balance profit and cost" is used throughout the manuscript. This is not optimal wording. Profit is the total revenue minus costs. Balance gain and cost is more precise.

In the abstract and text "long distance cell-to-cell communication". Cell-to-cell is mainly used for short-range communication and to include this into long-range signaling is confusing.

Page 3 line 67 – 78. The possible involvement of CLV2 in autoregulation is left out, there is at least one publication.

Page 3 line 92- 94 is unclear. "application of analogy" does not make sense

Page 4 line 124 " and same nodule size was continuously measured" . Please describe what was measured and compared. The legend make sense line 124 does not.

Page 5 line 133. In genetic terminology "suppress" is often used for a second mutation suppressing the effect of a previously induced mutation. ... nrsym1 mutations eliminates the pleiotrophic.....

Page 5 line 139 – 141. The sentence starting "Although -.... is convoluted, please rephrase

Page 8 line 238. .. "constitutive expression" which promoter was used?

Page 8 line 241. In contrast is better than "Alternatively"

Reviewer #2 (Remarks to the Author):

The manuscript of Nishida et al. reports the identification of a new major molecular player involved in the nitrate-dependent pleiotropic control of root nodule symbiosis. Root nodule symbiosis is tightly regulated by several mechanisms. This is critical for the performance of legumes and whereas the deciphering of the autoregulation (AON) via long distance signals has seen large progress during the recent years, the nitrate-induced control has so far remained largely enigmatic. The results presented are well described and analyzed. They are novel and add important new knowledge on the regulation of nodulation.

Based on the isolation of two EMS-mutants in *Lotus japonica*, which formed nodules in the presence of high concentration of nitrate in contrary to the wild type, the authors show that the *nrsym1* mutation suppresses the pleiotropic nitrate-induced inhibition of root nodule symbiosis and that the loss of this regulation impaired plant growth in the presence of nitrate and as well when both nitrate and rhizobia are present. For both mutants, they identified by genome sequencing a point mutation in a gene which belongs to the RWP-RK transcription factor family. Member of this gene family are major players of nitrate-regulated processes in *Arabidopsis* and *Chlamydomonas*. The authors then used a grafting strategy to show the possible interplay of nitrate-regulated nodulation and AON. Whereas the *NRSYM1* mutation affects all nitrate-regulated aspects of nodule development, the *har1* mutation is involved only in the regulation of the nodule number downstream of the *NRSYM* signaling pathway. In addition, they showed that the root-derived peptides CLE-RS1 and CLE-RS2 act on nodule number and that CLE-RS2 is specifically involved in the regulation of nodule number by nitrate.

Indeed nitrate regulation of the expression of the nodule specific CLE-RS2 peptide and the nitrate assimilation genes *NIA* and *NIR1* are altered in the *nrsym1-1* mutant and the *NRSYM1* protein binds to predicted cis-elements in the promoter of CLE-RS2 and *NIA*. In addition, a nitrate- supply depending accumulation of *NRSYMS1* in the nucleus was demonstrated.

Thus, similar as its homologs in *Arabidopsis thaliana*, the non-symbiotic NIN Like protein *NRSYM1* acts

on the nitrate-induced expression of nitrate assimilation enzymes and also shares the post-translational regulation in response to nitrate with the Arabidopsis homolog. In addition, NRSYM1 regulates the expression of CLE-RS2, a key player for the regulation of nodule number.

I have several comments:

1) NRSYM1 nuclear accumulation: It is proposed that NRSYM1 is activated by a nitrate-dependent nuclear retention mechanism. While this is certainly a tempting hypothesis based on results on Arabidopsis homologs from the RWP-RK transcription factor family, the results presented do not allow to firmly conclude this (abstract line 42/43). In Fig 7D, it is shown that NRSYM1 is localized in the nuclei after a 24h nitrate treatment, and that the protein is barely detectable in the nuclei in the absence of nitrate. This clearly indicates a post-transcriptional control mechanism, but further data are needed to demonstrate that the mechanism is identical to the nitrate-regulated nuclear retention proposed to regulate AtNLP7. A by far shorter response kinetics after nitrate addition and the use of drugs inhibiting nuclear export and import would allow suggesting that the nuclear accumulation of NRSYM1 is regulated by nuclear retention.

2) Except for Figure 1a and b (nodule numbers in wt and mutant under different nitrate availabilities) all data are shown for one allelic mutant. However EMS mutants contain hundreds of mutations and thus either several allelic mutants or complemented lines have to be analyzed in order to confirm that the phenotype is indeed due to one mutation. This has been done only for the trait "nodule number". Data should be added at least to Figures 1c-g and 6 a-c.

2) Expression pattern of NRSYM1 was analyzed using RT-PCR and Promoter-GUS expression in roots and nodules (lines 182-187). NRSYM1 regulates the expression of enzymes involved in nitrate assimilation. Is nitrate assimilation taking place in roots and shoots in legumes? Data for the expression pattern in shoots would give further information for the biological role of NRSYM1

3) Nitrate-dependent role of NRSYM: a) NRSYM effect on gene expression (Figure 6) and its binding to target sequences (Figure 7) was studied in response to a 24h nitrate treatment. In Arabidopsis NLPs are involved in very rapid responses to nitrate (within minutes). Plant metabolism has changed completely after 24h of nitrate-resupply and other metabolites might regulate NRSYM1. In order to strengthen the results, shorter time points are required. b) In addition to nitrate, also ammonium or other N sources do repress nodulation. A comparison between nitrate treatment and another Nitrogen source treatment would allow concluding if the observed differences between mutants and WT are indeed due to nitrate itself. c) Whereas nodulation is repressed by nitrate concentration in the mM range, primary nitrate response in Arabidopsis is triggered by far lower concentrations of nitrate (μM). Here it is shown that NRSYM1 regulates the expression of nitrate assimilation pathway enzymes and CLE-RS2 peptide expression, the later involved in the regulation of nodule number. In all experiments 10mM nitrate was used, but it would be interesting to show if concentrations that do not inhibit nodulation, would induce the expression of nitrate-responsive genes in a NRSYM1 dependent manner.

4) Target genes of NRSYM1: Several genes that are regulated by nitrate have been presented in this MS, but no full genome approach (RNAseq or ChIPseq) has been used to obtain a full picture of NRSYM1 target genes. This might be not essential; however, the number of NRSYM1-regulated genes in response to nitrate should be extended in order to cover some essential genes in the context of nodulation: NIN expression is repressed by nitrate. Is this repression dependent on NRSYM1? And also CEP expression is induced by low Nitrogen availability – are they directly regulated by NRSYM1? NIA expression is reduced in the mutant background, but no data are given for the hairy root overexpressing approach; is the expression of CLE-RS1 unchanged in the mutant background, as it is the case for overexpression?

5) CLE-RS2 : The authors included data on loss-of-function CLE mutants into the manuscript. Indeed they analyzed the relationship between AON and the nitrate effect on nodulation. However, the CLE loss of function genotypes are presented in the MS before the reader knows that CLE-RS2 is a target gene of NRSYM1. I would suggest to either presenting this data as supplementary data, or after figure 7. Data on CLE-loss-of-function genotypes are shown only for nodulation related traits. What is the

role of CLE RS2 for other traits (root growth)?

6) The discussion reads well, but I was missing several elements: (i) Very recently the regulation of NLP-dependent nitrate signaling by CDK has been shown (Liu et al 2017), (ii) Nitrate transporters are acting as nitrate receptors in Arabidopsis. Some evidences are also known from legumes that NRT proteins sense nitrate. This should be discussed (Badchi et al. 2012, Pellizaro et al. 2014). (iii) several other LjNLP do exist, one which is homologous to NRSYM1 and in the same clade as AtNLP7. Any knowledge on these genes?

Minor comments:

a) Lines 167-168: The nucleotide substitutions are given. It would be useful for the reader to know at the same time the changes in the amino acids.

b) Lines 256-259: NRSYMS1 binding sites: Please give the sequences of the binding sites and indicate the differences between NRE and NBS, for example in a supplementary Figure.

c) Lines 354/355: please precise: Arabidopsis NLP6/7 instead of NLPs

d) Figure 1b: In respect to the nodule numbers (>0.5mm) wt and mutants show a similar reduction when supplied with 50mM nitrate. This has not been commented in the text.

e) Figure 2 b and c should be presented in the same manner in order to allow comparison easily.

Reviewer #3 (Remarks to the Author):

The paper "A NIN-LIKE PROTEIN mediates nitrate-induced control of root nodule symbiosis in *Lotus japonicus*" by Nishida et al. describes the role of a NIN-like protein (NLP) in the inhibition of nodule numbers by nitrate.

In the past, the nitrate-sensing mechanism of plants has been mainly described in *Arabidopsis thaliana*, where NLPs function as master regulators of the nitrate response (Castaings et al., 2009; Konishi and Yanagisawa, 2010, 2011a, b; Konishi and Yanagisawa, 2013; Marchive et al., 2013). It has also been reported that NIN (a protein with RWP-RK domain as DNA binding domain), targets the promoters of CLE-RS1 and CLE-RS2 for regulating nodule numbers via a systemic inhibitory mechanism, AON (Soyano et al., 2014). However, the role of *Lotus* NLPs for regulating nodule numbers by nitrate sensing remained unknown.

This paper describes a *L. japonicus* mutant (*nrsym1*) which is impaired in suppressing nodulation upon high nitrate concentrations. Key processes in nodulation such as: rhizobial infection, nodule growth, and nitrogen fixation activity were not reduced in the *nrsym1-1* mutants when nitrate was added. Suggesting that NRSYM1 plays a role in the regulation of root nodule symbiosis upon nitrate availability.

The mutated gene responsible for the phenotype was identified by a map-based cloning approach as a NIN-like protein, which inhibits nodulation upon nitrate treatments. Mutants were rescued with the complementation construct, indicating that the *nrsym1* allele contributes to the phenotype observed. Phylogenetic analysis suggested that NRSYM1 represents member of a NIN-like protein clade previously reported in *Arabidopsis* as nitrate sensing proteins. Similar to the previously reported nitrate-sensitive NLPs, NRSYM1 also re-localized to the nucleus when nitrate was applied. So far, in *Lotus japonicus* it has not been possible to differentiate the role of CLE-RS1 from the one of CLE-RS2. Here, the authors engineered CRISPR/Cas9 mutants of both peptides, which allowed them to conclude that only CLE-RS2 is required for the inhibition of nodulation by nitrate. They propose a mechanism by which NRSYM1 directly regulates CLE-RS2 and after its activation, the signaling pathway converges to the one of AON. These findings provide insights into how plants coordinate responses to nitrate availability, linking nitrate sensing with the regulation of nodules.

Major comments

- the authors claim that the transcription factor NRSYM1 binds to the promoters of and activates CLE_RS2. However the evidence is insufficient and the NIN vs. NLP data are somewhat conflicting. DNA-protein interaction data and transactivation data need to be provided to further resolve this issue.
- The authors suggest that NRSYM1 directly regulates CLE-RS2, with two experiments a) qPCR data and b) ChIP. However, the region where NRSYM1 binds to CLE-RS2, should be further delimited and tested for specificity. They should also provide direct evidence for transactivation of CLE-RS2 by NRSYM1.
- For the ChIP assays, the authors mention that "Although CLE-RS2 promoter region 3 and 4 contain NBS, NRSYM1 did not seem to bind to the regions, implying that the binding sites for NRSYM1 and NIN are not completely conserved". However, in Soyano et al. 2014 it was reported that NIN binds to CLE-RS2 also in a region around -4000bp, the same where NRSYM1 binds. How is this binding compared to the one of NIN? And which one has a higher affinity for the promoter?
- The authors suggest that the AtNLP6/7 nitrate-response is conserved in NRSYM1. If this is the case, then the *nrsym1* could be complemented with an AtNLP6/7, or a *nlp6/7* with NRSYM1. This will also help with inferences on the protein domains responsible for binding and nitrate sensing.
- The nitrate-dependent nuclear retention evidence for NRSYM1 needs quantitative data, specially because in nitrate absence there are some nuclei containing the fusion(?) construct.
- The cell area displayed in figure 2 (1,500 square um per infected cell) should be double checked. In the images the variation among the cells appears larger than suggested by the error bars. How many cells per section were analyzed and was there a size selection done? Please clarify in materials and methods or figure legend.
- The authors mention that "NRSYM1 has a role regulating the expression of general nitrate-responsive genes involved in nitrate assimilation" (345-347). As also implied in their final model. However, this is only confirmed by qPCR and ChIP experiments. Binding and transactivation of LjNIR1 by NRSYM1 is required.
- According to Nature Communications publishing rules all t-tests should provide t-values and degrees of freedom, here missing.
- Figures 1b, 4e do not contain any statistical analysis.
- Authors claim that *nrsym1* is a recessive mutant allele; however, there is no segregation analysis provided to support that claim.

Minor comments

- 248-250 Which is this observation? Do they mean (Fig. 6).
"An observation of the NRSYM1-dependent expression of some nitrate-inducible genes led us to postulate that NRSYM1, an NLP transcription factor, directly regulates the expression of these genes"
- Fig.2 description: typo: "for (a),"
- NRSYM1, located in chr5.CM0148.170.r2.a, was already named in Suzuki et al., 2013 as LjNLP4.

Castaings, L., Camargo, A., Pocholle, D., Gaudon, V., Texier, Y., Boutet-Mercey, S., Taconnat, L., Renou, J.P., Daniel-Vedele, F., and Fernandez, E. (2009). The nodule inception-like protein 7 modulates nitrate sensing and metabolism in Arabidopsis. *The Plant Journal* 57, 426-435.

Konishi, M., and Yanagisawa, S. (2010). Identification of a nitrate-responsive cis-element in the Arabidopsis NIR1 promoter defines the presence of multiple cis-regulatory elements for nitrogen response. *Plant J* 63, 269-282.

Konishi, M., and Yanagisawa, S. (2011a). The regulatory region controlling the nitrate-responsive expression of a nitrate reductase gene, NIA1, in Arabidopsis. *Plant & cell physiology* 52, 824-836.

Konishi, M., and Yanagisawa, S. (2011b). Roles of the transcriptional regulation mediated by the nitrate-responsive cis-element in higher plants. *Biochemical and biophysical research communications* 411, 708-713.

Konishi, M., and Yanagisawa, S. (2013). Arabidopsis NIN-like transcription factors have a central role in nitrate signalling. *Nature communications* 4, 1617.

Marchive, C., Roudier, F., Castaings, L., Bréhaut, V., Blondet, E., Colot, V., Meyer, C., and Krapp, A. (2013). Nuclear retention of the transcription factor NLP7 orchestrates the early response to nitrate in plants. *Nature communications* 4, 1713.

Soyano, T., Hirakawa, H., Sato, S., Hayashi, M., and Kawaguchi, M. (2014). NODULE INCEPTION creates a long-distance negative feedback loop involved in homeostatic regulation of nodule organ production. *Proceedings of the National Academy of Sciences* 111, 14607-14612.

Response to Reviewer #1

Experimental work:

The genetic inheritance of the *nrsym1* mutation is not included although a map based cloning approach was used. Is the mutation recessive? There should be segregation data available from the mapping. Please also include references presenting the MG20 and Gifu accessions as model legumes and their use as mapping partners in order to inform readers outside the field.

We added information to the Results section about the segregation ratio of the *nrsym1* mutation using an F2 population derived from a cross between *nrsym1* and WT (MG-20) plants. Our data indicate that the mutation is inherited as a recessive trait. We added two references for Gifu and MG20 (Handberg and Stougaard, Plant J, 1992; Kawaguchi, J Plant Res, 2000).

The CRISPR/Cas generated mutations of CLE-RS1 and CLE-RS2 are not clearly described. Which region of the genes were targeted and were possible off-targets in other CLE genes considered in the design? How were the mutants characterized molecularly to ensure they were not chimeric? Was segregation of the mutations analyzed? Were the mutants complemented to make sure the observed effects were due to inactivation of the CLE-RS1 and CLE-RS2 genes and no other members of the large CLE family that may be off-targets? These questions are important for the conclusions drawn.

We added more details to the figure relevant to the *cle-rs1* and *-rs2* mutants created by the CRISPR/Cas9 system (Supplementary Fig. 9a) and added information about the CRISPR lines in the Results section. The nucleotide sequences that were used to design the gRNA are shown in bold (Supplementary Fig. 9a). Among the *LjCLE* genes, *LjCLE49* and *CLE-RS3* were predicted as potential off-target candidates in the CRISPR-P program (<http://cbi.hzau.edu.cn/crispr/>), although their off-target scores were fairly low. We sequenced the *LjCLE49* and *CLE-RS3* genes from the *cle-rs1* #16, *cle-rs2* #2, *cle-rs2* #5 lines and confirmed that the two *CLE* genes were unaffected in these plants. Each T0 generation of the CRISPR lines already had homozygous indel mutations as shown in Supplementary Fig. 9a, and no other mutations in *CLE-RS1* or *-RS2* were detected. Thus, there were no chimeric mutations present in the three lines. For phenotypic analyses, we used the T2 generation, where we again sequenced the *CLE-RS1* and *-RS2* genes and confirmed that the respective homozygous

mutations were fixed. We tried to examine the segregation ratio of nodulation phenotypes in *cle-rs1* #16 (increased number of nodules in combination with *cle-rs2*) using the seeds derived from *CLE-RS1/cle-rs1 cle-rs2/cle-rs2* plants and *cle-rs2* #5 (unaffected number of nodules in the presence of high nitrate) using the seeds derived from *CLE-RS1/cle-rs1 CLE-RS2/cle-rs2* plants. The nodulation phenotypes did not segregate in a typical recessive manner. Because the nodulation phenotypes of *cle-rs1 -rs2* double and *cle-rs2* single mutants are relatively weak, it may be difficult to definitively select mutants using indicators of such a phenotype from some segregating populations. The introduction of the *CLE-RS1* genomic fragments into *cle-rs1 -rs2* double mutants rescued their nodulation phenotype (Supplementary Fig. 9b). For *cle-rs2*, we obtained two independent lines that exhibited almost identical phenotypes. Thus, we concluded that the observed phenotype was due to inactivation of *CLE-RS2*.

ChIP was used to show in vivo binding of NRSYM1 to region 3 of CLE-RS2 promoter in order to support the NRSYM1-CLE-RS2-HAR1 pathway. Any results for the importance of NRSYM1 binding from deletion of region 3 in the CLE-RS2 promoter?

Perhaps you mean region 1 of the *CLE-RS2* promoter. The results of EMSA and promoter-GUS analyses for which we mutated region 1 of the *CLE-RS2* promoter indicated that NRSYM1 binds to region 1 of the *CLE-RS2* promoter (Fig. 6d-f; Supplementary Fig. 8b,d). This result supports the ChIP results.

Presentation:

The wording is not always optimal. Examples are listed below but the list is probably not complete.

Thank you very much for your helpful comments to improve the wording. We revised our manuscript following your suggestions.

The biological processes that are central for the manuscript are not well-presented in the abstract “ To balance the profits and costs associated with the symbiosis, plants have developed two major strategies; plants can maintain nodule number and can stop the symbiosis upon nitrogen availability in the soil.” This statement is not incorrect but the meaning is unclear, even for experts. There are many ways to rephrase the sentence and I can suggesttwo strategies for adapting to nitrogen availability in the soil; plants can

regulate nodule numbers and/or stop development or function of processes central for symbiosis. The point is to be precise so readers outside the field understand.

We rephrased the sentence as you suggested.

“Balance profit and cost” is used throughout the manuscript. This is not optimal wording. Profit is the total revenue minus costs. Balance gain and cost is more precise.

We used “gain” instead of “profit” throughout the revised manuscript

In the abstract and text “long distance cell-to-cell communication”. Cell-to-cell is mainly used for short-range communication and to include this into long-range signaling is confusing.

We used “long-range signaling” instead of “long distance cell-to-cell communication”.

Page 3 line 67 – 78. The possible involvement of CLV2 in autoregulation is left out, there is at least one publication.

We provided a reference to a CLV2 paper (Krusell et al., Plant J, 2011).

Page 3 line 92- 94 is unclear. “application of analogy” does not make sense

We rephrased the sentence.

Page 4 line 124 “ and same nodule size was continuously measured” . Please describe what was measured and compared. The legend make sense line 124 does not.

This sentence was rewritten in the revised manuscript.

Page 5 line 133. In genetic terminology “suppress” is often used for a second mutation suppressing the effect of a previously induced mutation. ... nrsym1 mutations eliminates the pleiotrophic.....

We replaced “eliminate” with “suppress”.

Page 5 line 139 – 141. The sentence starting “Although -.... is convoluted, please rephrase

We rephrased the sentence.

Page 8 line 238. .. “constitutive expression” which promoter was used?

Information about the promoter was added.

Page 8 line 241. In contrast is better than “Alternatively”

We used “In contrast” instead of “Alternatively”.

Response to Reviewer #2

1) NRSYM1 nuclear accumulation: It is proposed that NRSYM1 is activated by a nitrate-dependent nuclear retention mechanism. While this is certainly a tempting hypothesis based on results on Arabidopsis homologs from the RWP-RK transcription factor family, the results presented do not allow to firmly concluded this (abstract line 42/43). In Fig 7D, it is shown that NRSYM1 is localized in the nuclei after a 24h nitrate treatment, and that the protein is barely detectable in the nuclei in the absence of nitrate. This clearly indicates a post-transcriptional control mechanism, but further data are needed to demonstrate that the mechanism is identical to the nitrate-regulated nuclear retention proposed to regulate AtNLP7. A by far shorter response kinetics after nitrate addition and the use of drugs inhibiting nuclear export and import would allow suggesting that the nuclear accumulation of NRSYM1 is regulated by nuclear retention.

A new figure (Fig. 8) shows a shorter response time (20 min after nitrate treatment) for NRSYM1 nuclear accumulation. We also show that nuclear accumulation was reversible when nitrate was withdrawn. The effect of leptomycin B (a nuclear export inhibitor) was examined, but we could not find an effective drug to serve as a nuclear import inhibitor. We quantified the percentage of NRSYM1-accumulating nuclei in respective growth conditions. Based on these results, we believe that the statement “NRSYM1 is activated by a nitrate-dependent nuclear retention mechanism” is supported.

2) Except for Figure 1a and b (nodule numbers in wt and mutant under different nitrate availabilities) all data are shown for one allelic mutant. However EMS mutants contain hundreds of mutations and thus either several allelic mutants or complemented lines have to be analyzed in order to confirm that the phenotype is indeed due to one mutation. This has been done only for the trait “nodule number”. Data should be added at least to Figures 1c-g and 6 a-c.

Results from phenotypic analyses of *nrsym1-2* (Supplementary Fig. 1a-e) and gene expression

analysis of *nrsym1-2* (Supplementary Fig. 7a-c) suggest that the *nrsym1-2* mutants basically have the same defects as *nrsym1-1*.

2) Expression pattern of NRSYM1 was analyzed using RT-PCR and Promoter-GUS expression in roots and nodules (lines 182-187) . NRSYMS1 regulates the expression of enzymes involved in nitrate assimilation. Is nitrate assimilation taking place in roots and shoots in legumes ? Data for the expression pattern in shoots would give further information for the biological role of NRSYM1

We examined *NRSYM1* expression in aboveground organs such as flowers, leaves, stems and shoot apices (Supplementary Fig. 6a) and found that *NRSYM1* was expressed in all organs examined. It is previously reported that in *L. japonicus* nitrate assimilation occurs predominantly in roots (Prosser et al., Planta, 2006). *LjNIA* and *LjNIR1* expression was induced by nitrate in leaves (Supplementary Fig. 7j,k). Although *LjNIA* and *LjNIR1* induction by nitrate was dependent on NRSYM1 in roots (Fig. 5b,c), NRSYM1 was not required to induce expression of the two genes in leaves (Supplementary Fig. 7j,k).

3) Nitrate-dependent role of NRSYM: a) NRSYMS effect on gene expression (Figure 6) and its binding to target sequences (Figure 7) was studied in response to a 24h nitrate treatment. In Arabidopsis NLPs are involved in very rapid responses to nitrate (within minutes). Plant metabolism has changed completely after 24h of nitrate-resupply and other metabolites might regulate NRSYM1. In order to strengthen the results, shorter time points are required. b) In addition to nitrate, also ammonium or other N sources do repress nodulation. A comparison between nitrate treatment and another Nitrogen source treatment would allow concluding if the observed differences between mutants and WT are indeed due to nitrate itself . c) Whereas nodulation is repressed by nitrate concentration in the mM range, primary nitrate response in Arabidopsis is triggered by far lower concentrations of nitrate (μM). Here it is shown that NRSYM1 regulates the expression of nitrate assimilation pathway enzymes and CLE- RS2 peptide expression, the later involved in the regulation of nodule number. In all experiments 10mM nitrate was used, but it would be interesting to show if concentrations that do not inhibit nodulation, would induce the expression of nitrate-responsive genes in a NRSYM1 dependent manner.

We examined nitrate-inducible gene expression at shorter time points (Supplementary Fig. 7d-f), and found that the timing for the induction of *CLE-RS2* expression was slower than for other

genes involved in the nitrate assimilation process. We also examined the effects of ammonium on nodulation (Supplementary Fig. 1f) and found that ammonium eliminated nodulation in *nrsym1* as well as WT. This result suggests that NRSYM1 is not involved in the ammonium-induced control of nodulation. Application of 200 μ M nitrate induced *CLE-RS2*, *LjNIA* and *LjNIR1* expression in an NRSYM1-dependent manner (Supplementary Fig. 7g-i). The activation of *CLE-RS2* by a low nitrate concentration seemed to be insufficient for inhibiting nodulation. There might be an unidentified mechanism in response to a high nitrate concentration that serves to inhibit nodulation in parallel or downstream of *CLE-RS2* activation.

4) Target genes of NRSYM1: Several genes that are regulated by nitrate have been presented in this MS, but no full genome approach (RNAseq or ChIPseq) has been used to obtain a full picture of NRSYM1 target genes. This might be not essential; however, the number of NRSYM1-regulated genes in response to nitrate should be extended in order to cover some essential genes in the context of nodulation: NIN expression is repressed by nitrate. Is this repression dependent on NRSYM1? And also CEP expression is induced by low Nitrogen availability – are they directly regulated by NRSYM1? NIA expression is reduced in the mutant background, but no data are given for the hairy root overexpressing approach; is the expression of CLE-RS1 unchanged in the mutant background, as it is the case for overexpression?

Indeed, an important next research project will be to identify NRSYM1 target genes other than *CLE-RS2* during nodulation. The expression of *NIN* tended to be down-regulated by nitrate in WT, and the effect was not observed in *nrsym1* (Supplementary Fig. 7l). In contrast to *Arabidopsis*, the relationship between *CEP* expression and nitrogen starvation is unknown in *L. japonicus* currently. Also, in this paper we are focusing on the signaling in response to sufficient nitrogen. The signaling in response to nitrogen deficiency and its implication for NRSYM1 will be the subject for another research project, in which we will look for *LjCEP* genes whose expression is induced by nitrogen starvation. Data for *LjNIA* expression in NRSYM1-OX roots and *CLE-RS1* expression in *nrsym1* was added to the revised manuscript (Fig. 5d,f).

5) CLE-RS2 : The authors included data on loss-of-function CLE mutants into the manuscript. Indeed they analyzed the relationship between AON and the nitrate effect on nodulation. However, the CLE loss of function genotypes are presented in the MS before the reader knows that CLE-RS2 is a target gene of NRSYM1. I would suggest to either

presenting this data as supplementary data, or after figure 7. Data on CLE-loss-of-function genotypes are shown only for nodulation related traits. What is the role of CLE RS2 for other traits (root growth)?

We moved the Figure showing the loss-of-function phenotype of *CLE-RS1* and *-RS2* to a position after the Figure showing that *CLE-RS2* is a target of NRSYM1. In our preliminary experiment, no obvious non-symbiotic defects were observed in the *cle-rs2* mutants, but this conclusion needs to be more carefully verified in several growth conditions as an independent study.

6) The discussion reads well, but I was missing several elements: (i) Very recently the regulation of NLP-dependent nitrate signaling by CDK has been shown (Liu et al 2017), (ii) Nitrate transporters are acting as nitrate receptors in Arabidopsis. Some evidences are also known from legumes that NRT proteins sense nitrate. This should be discussed (Badchi et al. 2012, Pellizaro et al. 2014). (iii) several other LjNLP do exist, one which is homologous to NRSYM1 and in the same clade as AtNLP7. Any knowledge on these genes?

We discussed these points in the Discussion section as you suggested. Suzuki et al. reported that Lj1g3v2295200 (LjNLP1) can bind to NRE and respond to nitrate (Suzuki et al., Plant Signal Behav, 2013). There is currently no information on other LjNLPs, including Lj2g3v0381780.

Minor comments:

a) Lines 167-168: The nucleotide substitutions are given. It would be useful for the reader to know at the same time the changes in the amino acids.

We added information about the amino acids that were substituted by the mutations.

b) Lines 256-259: NRSYMS1 binding sites: Please give the sequences of the binding sites and indicate the differences between NRE and NBS , for example in a supplementary Figure.

We added the sequences in Supplementary Fig. 8b.

c) Lines 354/355: please precise: Arabidopsis NLP6/7 instead of NLPs

We rephrased this identification as you suggested.

d) Figure 1b: In respect to the nodule numbers (>0.5mm) wt and mutants show a similar reduction when supplied with 50mM nitrate. This has not been commented in the text.

We commented on this point in the revised text.

e) Figure 2 b and c should be presented in the same manner in order to allow comparison easily.

Fig. 2b and c were revised so as to be presented consistently.

Response to Reviewer #3

Major comments

• the authors claim that the transcription factor NRSYM1 binds to the promoters of and activates CLE_RS2. However the evidence is insufficient and the NIN vs. NLP data are somewhat conflicting. DNA-protein interaction data and transactivation data need to be provided to further resolve this issue.

The results of EMSA and promoter-GUS assays showed that NRSYM1 can activate *CLE-RS2* through direct binding to region 1 of the *CLE-RS2* promoter (Fig. 6d-f; Supplementary Fig. 8b,d), a result that is consistent with the ChIP results. A transactivation study using *Nicotiana benthamiana* did not work probably because endogenous NLPs in *N. benthamiana* might affect reporter gene expression. As an alternative and more reliable method, we used a promoter-GUS assay using *L. japonicus* hairy roots and showed the NRSYM1-dependent activation of *CLE-RS2*.

• The authors suggest that NRSYM1 directly regulates CLE-RS2, with two experiments a) qPCR data and b) ChIP. However, the region where NRSYM1 binds to CLE-RS2, should be further delimited and tested for specificity. They should also provide direct evidence for transactivation of CLE-RS2 by NRSYM1.

We delimited the region using EMSA. The length of DNA probes (47 or 48 bp) used in EMSA are shorter than the DNA fragments (133-211 bp) amplified in ChIP-qPCR. In EMSA and the promoter-GUS assay, we mutated region 1 of the *CLE-RS2* promoter to test the specificity.

• For the ChIP assays, the authors mention that “Although CLE-RS2 promoter region 3 and 4 contain NBS, NRSYM1 did not seem to bind to the regions, implying that the binding

sites for NRSYM1 and NIN are not completely conserved". However, in Soyano et al. 2014 it was reported that NIN binds to CLE-RS2 also in a region around -4000bp, the same where NRSYM1 binds. How is this binding compared to the one of NIN? And which one has a higher affinity for the promoter?

In the ChIP assays, we did not detect NRSYM1 binding to regions 3 and 4 of the *CLE-RS2* promoter (Fig. 6a,b). In contrast, the EMSA result suggests that NRSYM1 can bind to regions 3 and 4 as well as region 1 (Fig. 6d). The difference in these results may be related to differences between *in vivo* and *in vitro* experiments. A competition assay showed that NRSYM1 binding to region 1 has the strongest affinity among the three regions tested (Fig. 6e). NRSYM1 and NIN bind to same area of region 1 in the *CLE-RS2* promoter. We compared the strength of NRSYM1 binding to the region with that of NIN and found that NRSYM1 had a higher affinity for the region than NIN (Supplementary Fig. 8c).

• **The authors suggest that the AtNLP6/7 nitrate-response is conserved in NRSYM1. If this is the case, then the *nrsym1* could be complemented with an AtNLP6/7, or a *nlp6/7* with NRSYM1. This will also help with inferences on the protein domains responsible for binding and nitrate sensing.**

Constitutive expression of *AtNLP6* or *AtNLP7* by the *LjUBQ* promoter partially rescued the *nrsym1* nodulation phenotype (Supplementary Fig. 4), suggesting that the functions of NRSYM1 and AtNLP6/7 are partially conserved.

• **The nitrate-dependent nuclear retention evidence for NRSYM1 needs quantitative data, specially because in nitrate absence there are some nuclei containing the fusion(?) construct.**

In a new figure (Fig. 8), we provided quantitative data (Fig. 8f) from a new data set in which the percentage of NRSYM1-accumulating nuclei was calculated.

• **The cell area displayed in figure 2 (1,500 square um per infected cell) should be double checked. In the images the variation among the cells appears larger than suggested by the error bars. How many cells per section were analyzed and was there a size selection done? Please clarify in materials and methods or figure legend.**

We added this information to the figure legends. To calculate cell area, the cell area of all infected and uninfected cells located within the inner region of nodule parenchyma were

measured and averaged (170-637 cells per nodule section). Using the obtained average cell area in respective nodule sections, we calculated the average cell area per nodule section (n = 4–5 nodules section). No size selection was done.

• The authors mention that “NRSYM1 has a role regulating the expression of general nitrate-responsive genes involved in nitrate assimilation” (345-347). As also implied in their final model. However, this is only confirmed by qPCR and ChIP experiments. Binding and transactivation of LjNIR1 by NRSYM1 is required.

The results of EMSA and promoter-GUS assays showed that NRSYM1 can activate *LjNIR1* through direct binding to region 3 of the *LjNIR1* promoter (Fig. 6d-f; Supplementary Fig. 8b,d). This result is consistent with the ChIP result.

• According to Nature Communications publishing rules all t-tests should provide t-values and degrees of freedom, here missing.

We provided t-values and degrees of freedom for all figures in the revised manuscript.

• Figures 1b, 4e do not contain any statistical analysis.

We performed a statistical analysis of the data presented in Fig. 1b and 4e. In Fig. 4e, we provided a new data set in the revised manuscript.

• Authors claim that *nrsym1* is a recessive mutant allele; however, there is no segregation analysis provided to support that claim.

We added information to the Results section about the segregation ratio of the *nrsym1* mutation using an F2 population derived from a cross between *nrsym1* and WT (MG-20) plants that indicates the mutation is inherited as a recessive trait.

Minor comments

• 248-250 Which is this observation? Do they mean (Fig. 6).

“An observation of the NRSYM1-dependent expression of some nitrate-inducible genes led us to postulate that NRSYM1, an NLP transcription factor, directly regulates the expression of these genes”

We are referring to a new figure (Fig. 5) in this sentence.

- **Fig.2 description: typo: “for (a),”**

Thank you, we corrected this typographical error.

- **NRSYM1, located in chr5.CM0148.170.r2.a, was already named in Suzuki et al., 2013 as LjNLP4.**

We indicated that NRSYM1 is the same as LjNLP4 in the revised text.

REVIEWERS' COMMENTS:

Reviewer #1 (Remarks to the Author):

The authors have addressed all the points raised by reviewers. New results have been added to support conclusions and improved experimental conditions have been implemented to refine some of the studies. This improved manuscript is an important contribution to our understanding of nitrogen regulation in developmental processes of plants and symbiosis in particular. The results are of general interest for a broad audience and the manuscript is well written and ready for publication.

Reviewer #2 (Remarks to the Author):

The authors provided important new additional data and clarified several points. Concerning the mechanism of NRSYM1 nuclear accumulation in response to nitrate (my comment 1) I have a comment to the newly added data.

Line 351ff and Figure 8e : In order to demonstrate that the accumulation of NRSYM1 in the nucleus in response to nitrate is due to an inhibition of nuclear export, the authors analyzed the effect of inhibiting export in conditions where NRSYM1 is in the processes of export and showed that NRSYM1 accumulation in the nucleus persists after transferring nitrate induced plants to N free medium in the presence of LMB. This result nicely demonstrates that that LMB-sensitive processes are involved in the diminution of nuclear accumulation after transfer to N free medium, but is not supporting that the accumulation is based on nitrate-regulated export inhibition. Other data must be shown to support clearly this mechanism.

Reviewer #3 (Remarks to the Author):

The authors have strongly improved the data support for their claim that NRSYM1 regulates the expression of both CLE-RS2 and NIR1 promoters via specific binding to their promoters. Competition analysis also improved both: specificity of DNA binding and evidence that NRSYM1 has a higher affinity for the region than NIN.

They also described the effect of NRSYM1 for the activation of both CLE-RS2 and NIR1 promoters with promoter:GUS analyses in *Lotus japonicus* roots , which are consistent with their EMSA and ChIP experiments.

The subcellular localization of NRSYM1, in conditions of starvation and uptake of nitrate, has been extensively improved. Authors provided support that NRSYM1 has a similar mechanism to the previously described in AtNLP7, with nitrate sensing function mediated by a nitrate-dependent nuclear retention mechanism and re-localization of the protein in the different conditions.

The partial complementation of the mutant phenotype with homologs from *Arabidopsis*, supports that the functions of NRSYM1 and AtNLP6/7 are conserved. The authors also incorporated evidence for the claim that the *nrsym1* mutation is inherited as a recessive trait.

Overall the new manuscript is greatly improved. I only have minor formal points:

- 1) Read Instructions to authors of *The Plant Cell* for the standard nomenclature of promoter: GUS fusions. It must be promoter:GUS and not promoter::GUS.
- 2) in figure 1d please change: "days after supplying with our without nitrate" with something grammatically and scientifically correct.

3) In Figure 8 please include keyword text directly into the panels, in addition to the a,b,c ... labels. This will provide easier access to the experimental details by the reader.

Response to Reviewer #2

The authors provided important new additional data and clarified several points.

Concerning the mechanism of NRSYM1 nuclear accumulation in response to nitrate (my comment 1) I have a comment to the newly added data.

Line 351ff and Figure 8e : In order to demonstrate that the accumulation of NRSYM1 in the nucleus in response to nitrate is due to an inhibition of nuclear export, the authors analyzed the effect of inhibiting export in conditions where NRSYM1 is in the processes of export and showed that NRSYM accumulation in the nucleus persists after transferring nitrate induced plants to N free medium in the presence of LMB. This result nicely demonstrates that that LMB-sensitive processes are involved in the diminution of nuclear accumulation after transfer to N free medium, but is not supporting that the accumulation is based on nitrate-regulated export inhibition. Other data must be shown to support clearly this mechanism.

Indeed, the current data is insufficient to support the point of NRSYM1 nuclear accumulation by nitrate-regulated export inhibition. Thus, we removed definitive claims and toned down them regarding NRSYM1 nuclear localization in Abstract, Results, Discussion, and Figure legends sections.

Response to Reviewer #3

Overall the new manuscript is greatly improved. I only have minor formal points:

1) Read Instructions to authors of The Plant Cell for the standard nomenclature of promoter: GUS fusions. It must be promoter:GUS and not promoter::GUS.

We replaced “:” with “::”.

2) in figure 1d please change: "days after supplying with our without nitrate" with something grammatically and scientifically correct.

We rephrased it. As the same expression was used in Supplementary Fig. 1b, it was also rephrased.

3) In Figure 8 please include keyword text directly into the panels, in addition to the a,b,c ... labels. This will provide easier access to the experimental details by the reader.

We included keywords to explain each panel.